# Ancient diversity in host-parasite interaction genes in a model parasitic nematode

Lewis Stevens [1] ✉, Isaac Martínez-Ugalde [2], Erna King [1], Martin Wagah[1], Dominic Absolon[1], Rowan Bancroft[2], Pablo Gonzalez de la Rosa[1], Jessica L. Hall[2], Manuela Kieninger [1], Agnieszka Kloch[3], Sarah Pelan[1], Elaine Robertson[4], Amy B. Pedersen[2], Cei Abreu-Goodger [2], Amy H. Buck [4] & Mark Blaxter [1] ✉

Host-parasite interactions exert strong selection pressures on the genomes of both host and parasite. These interactions can lead to negative frequency-dependent selection, a form of balancing selection that is hypothesised to explain the high levels of polymorphism seen in many host immune and parasite antigen loci. Here, we sequence the genomes of several individuals of *Heligmosomoides bakeri*, a model parasite of house mice, and *Heligmosomoides polygyrus*, a closely related parasite of wood mice. Although *H. bakeri* is commonly referred to as *H. polygyrus* in the literature, their genomes show levels of divergence that are consistent with at least a million years of independent evolution. The genomes of both species contain hyper-divergent haplotypes that are enriched for proteins that interact with the host immune response. Many of these haplotypes originated prior to the divergence between *H. bakeri* and *H. polygyrus*, suggesting that they have been maintained by long-term balancing selection. Together, our results suggest that the selection pressures exerted by the host immune response have played a key role in shaping patterns of genetic diversity in the genomes of parasitic nematodes.

The interaction between hosts and their parasites is a key driver of biological diversity. Two possible outcomes of host-parasite coevolution are the repeated fixation of advantageous alleles in the host and parasite, referred to as an 'evolutionary arms race', and the continuous fluctuation in allele frequencies over time, referred to as 'Red Queen' or 'trench warfare' dynamics[1–3]. Red Queen dynamics result from negative frequency-dependent selection, a form of balancing selection where rare alleles have a fitness advantage over common alleles, which leads to the maintenance of genetic diversity that would otherwise be lost due to genetic drift[2–4]. In hosts, immune-relevant regions believed to be evolving under balancing selection include the vertebrate major histocompatibility (MHC) loci[5] and disease-resistance loci (*R* genes) in plants[6]. In parasites, evidence for balanced loci has predominantly

been found in protozoans, such as in the human malaria parasite *Plasmodium falciparum*, where elevated diversity has been found in multiple antigenic genes expressed at the parasite surface[7–11]. In contrast, parasitism-relevant regions evolving under balancing selection in many other parasite groups, including in nematodes, are far less well-studied. Approximately 50% of all described nematode species are parasites, including many that cause important diseases in humans, livestock and plants[12,13]. However, identifying balanced loci in parasitic nematodes is complicated by high levels of standing genetic diversity, even in local populations[14,15]. Recent work in free-living *Caenorhabditis* nematodes found that their genomes contain hundreds of hyper-divergent haplotypes that are enriched in environmental response genes, including those with roles in pathogen defence[16–19]. Although

[1]Tree of Life, Wellcome Sanger Institute, Hinxton, UK. [2]Institute of Ecology and Evolution, School of Biological Sciences, University of Edinburgh, Edinburgh, UK. [3]Faculty of Biology, University of Warsaw, Warsaw, Poland. [4]Institute of Immunology & Infection Research, School of Biological Sciences, University of Edinburgh, Edinburgh, UK. ✉e-mail: ls30@sanger.ac.uk; mb35@sanger.ac.uk

their origins are unknown, these haplotypes are hypothesised to be the product of long-term balancing selection[16,17]. A recent effort to generate a chromosome-level reference genome for a semi-inbred strain of *Haemonchus contortus*, an economically important nematode parasite of sheep, was complicated by high levels of haplotype diversity, suggesting that similar haplotypes may be present in parasite genomes[20]. However, the extent of these haplotypes in parasites, what genes they contain, and what relevance they have, if any, for host-parasite interaction is not yet known.

A key first step in the study of parasite genetics is the generation of high-quality reference genomes. In recent years, advances in genome sequencing technologies, particularly long-read technologies offered by Pacific Biosciences (PacBio) and Oxford Nanopore Technologies, and long-range technologies, such as chromosome-conformation capture (Hi-C), mean it is now possible to generate a high-quality, chromosome-level reference genome for almost any species[21]. Along with decreasing sequencing costs, these developments have led to the initiation of projects that aim to generate reference genomes for all species on Earth[22,23]. However, even with these advancements, generating high-quality reference genomes for parasitic nematodes remains challenging. Obtaining material for sequencing is often difficult because the larger, mature stages of many parasitic nematodes can only be sampled in vivo and are only accessible from the host post-mortem. In addition, standard long-read sequencing protocols typically require hundreds of nanograms of input DNA, far more than is present in most individual parasites. Furthermore, because nematode populations typically contain high levels of genetic diversity, pooling individuals prior to sequencing can lead to assembly artefacts[20,24]. As a result, chromosome-level reference genomes exist for only a handful of animal parasitic nematodes, primarily those of medical or veterinary importance where individuals are large or the species has been inbred[20,25–30].

*Heligmosomoides bakeri* is a parasitic nematode that naturally infects house mice (*Mus musculus*)[31]. The ability of *H. bakeri* to form chronic infections in mice[32], along with its close phylogenetic relationship to hookworm parasites of humans and livestock (Fig. 1a), has led to it becoming a valuable model for gastrointestinal nematode infection. Despite its status as a model organism, the taxonomy of *H. bakeri* has been a source of debate. The laboratory strain of *H. bakeri* is descended from a single isolation from a deer mouse (*Peromyscus maniculatus gambeli*) in California in 1950, which was subsequently provided to the Wellcome Foundation Laboratories in London before being distributed to other laboratories around the world[33]. This isolate was initially identified by morphology as *Nematosporoides dubius*[34], a common intestinal parasite of wood mice (*Apodemus sylvaticus*) in Europe[35]. *N. dubius* was later formally renamed *Heligmosomoides polygyrus*[35]. However, morphological comparisons found clear differences between the laboratory isolate and wild *H. polygyrus* isolates, leading to the two taxa being raised to subspecies (*H. polygyrus polygyrus* for the parasite of *A. sylvaticus*, and *H. polygyrus bakeri* for the laboratory isolate)[35]. In addition to morphological differences, *H. polygyrus* is not able to mature to adulthood in laboratory mice, and, although *H. bakeri* can mature in *Apodemus sylvaticus*, infections are short-lived unless the mice are treated with immunosuppressant drugs[36]. The two taxa were raised to full species by Cable et al. [37], who found substantial divergence between ribosomal and mitochondrial genes. However, this evidence was met with scepticism[38], and *H. bakeri* continues to be referred to as *H. polygyrus* in the immunological scientific literature.

Here, we sequenced the genomes of several individuals of *H. bakeri*, collected from a laboratory population of house mice (*M. musculus*), and *H. polygyrus*, collected from a wild population of wood mice (*A. sylvaticus*) in Scotland, using a recently developed low-input protocol. We combined our single nematode assemblies with Hi-C data to generate high-quality, chromosome-level reference genomes for both species. We found that their genomes show levels of divergence consistent with at least a million of years of independent evolution. In addition, we found that the *H. bakeri* genomes, which we expected to be highly inbred, contained hundreds of hyper-divergent haplotypes that are enriched in several protein families that have previously been implicated in nematode parasitism. Many of these haplotypes are shared between *H. bakeri* and *H. polygyrus* and levels of divergence suggest that they have been maintained since their last common ancestor by long-term balancing selection.

## Results

### Chromosome-level reference genomes for *H. bakeri* and *H. polygyrus*

We used the recently developed Picogram Input Multimodal Sequencing (PiMmS) protocol[39] to sequence the DNA of three *H. bakeri* individuals from a laboratory colony of house mice (*Mus musculus*) at the University of Edinburgh, and two *H. polygyrus* individuals collected from a population of wild wood mice (*Apodemus sylvaticus*) near Loanhead, Scotland. We generated between 6.2 and 33.9 Gb of PacBio HiFi data per individual, with read length N50s ranging from 9.6 to 11.1 kb (Supplementary Table 1). We assembled the genome of each individual independently and filtered out contigs originating from non-target organisms. Although most datasets showed minimal contamination, our *H. polygyrus* ngHelPoly2 preliminary assembly contained ~1.4 Gb of host sequence (*A. sylvaticus*) and ~9.7 Mb of sequence that corresponded to a *Giardia* sp. and a *Spironucleus* sp., two gastrointestinal diplomonad parasites of rodents (Supplementary Fig. 1). The other assemblies varied considerably in span, contiguity and completeness, largely driven by read coverage (Supplementary Table 1). For three individuals (nxHelBake1, nxHelBake2, and ngHelPoly1) that had over 25-fold read coverage after contaminant removal, we generated assemblies that spanned between 654.6 and 656.5 Mb, with contig N50s between 266 and 313 kb and high biological completeness (91.5 – 92.9% based on the Benchmarking Using Single Copy Orthologues, Nematoda odb10 dataset) (Fig. 1b). In contrast, the nxHelBake3 and ngHelPoly2 datasets, which had 8.8-fold and 14.2-fold coverage after contaminant removal, had reduced assembly spans (572.8 Mb and 620.0 Mb), were relatively fragmented (contig N50s of 102 kb and 137 kb), and had lower completeness (80.5% and 87.7%) (Supplementary Table 1). As expected, the *H. bakeri* individuals, which are descended from a line maintained in laboratory culture for over 70 years, were substantially less heterozygous than the wild-sourced *H. polygyrus* individuals (0.84-0.96% versus 1.67%, Supplementary Fig. 2).

To scaffold the single nematode assemblies into complete chromosomes, we sequenced two chromosome-conformation capture (Hi-C) libraries, derived from pools of individuals, to high coverage (147- to 183-fold). We independently scaffolded the most contiguous and complete assemblies for each species (nxHelBake1 and ngHelPoly1) and manually curated the resulting scaffolds by assessing congruence with Hi-C signal (Fig. 1c; Supplementary Fig. 3A). Our chromosome-level reference genomes for *H. bakeri* and *H. polygyrus* span 649.2 and 656.6 Mb with scaffold N50s of 110.8 and 107.4 Mb (Table 1). Consistent with karyotypes in related taxa[40,41], both genomes contain six multi-megabase scaffolds representing chromosomes (I–V, X), which comprise 98.6% and 96.2% of each assembly, along with a complete mitochondrial genome (Table 1). The six chromosomes of *H. bakeri* and *H. polygyrus* show strong conservation of gene content with the parasite *Haemonchus contortus*[20] (Fig. 1a; Supplementary Fig. 3C, D), including an X chromosome that is a fusion of the ancestral NigonN and NigonX elements[42] (Fig. 1d; Supplementary Fig. 3B). Although not phased, both reference genomes represent one haplotype, with corresponding alternate haplotype assemblies that span 304.5 (*H. bakeri*) and 623.8 Mb (*H. polygyrus*) (Supplementary Table 1). A majority of the chromosome ends in each reference genome are made up of tandem repeats of the nematode telomeric repeat sequence ([TTAGGC]n)

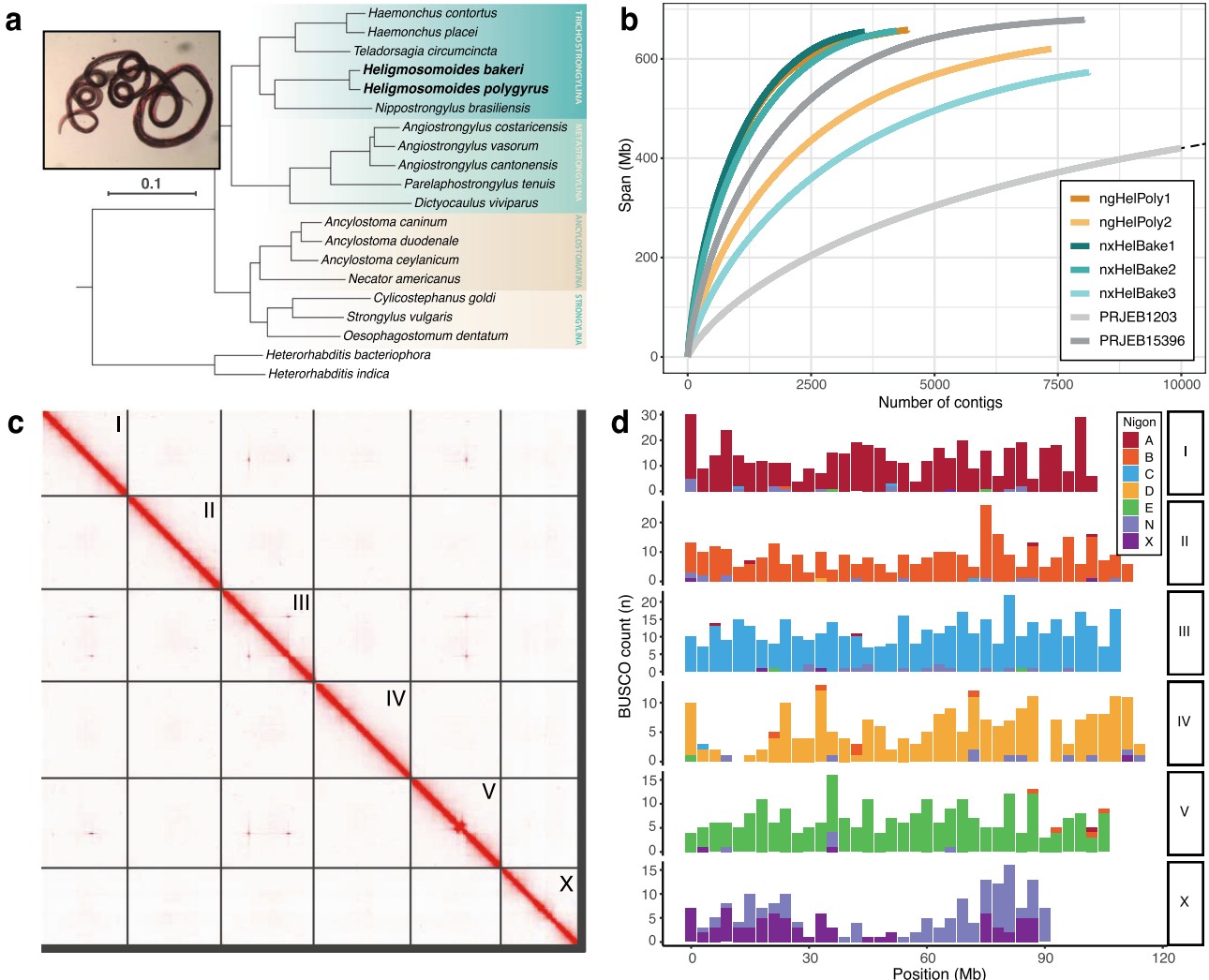

**Fig. 1 | Chromosome-level reference genomes for *Heligmosomoides bakeri* and *Heligmosomoides polygyrus*. a** The position of *H. bakeri* and *H. polygyrus* within Strongylida. Phylogeny was inferred using the amino acid sequences of 511 BUSCO genes that were present and single copy in at least 18 of the 20 species. *H. bakeri* and *H. polygyrus* are highlighted in bold. Suborders are indicated. Scale in sub-stitutions per site is shown. Inset: a *H. polygyrus* female. **b** Cumulative length plots showing contiguity of all single nematode assemblies and both existing *H. bakeri* reference genomes (PRJEB1203 and PRJEB15396). The dotted line indicates that the curve for the *H. bakeri* PRJEB1203 assembly extends beyond 10,000 contigs (the total number of contigs in the assembly is 20,569). **c** Hi-C contact map for the nxHelBake1.1 reference genome. Chromosome names are indicated. The Hi-C contact map for the ngHelPoly1.1 reference genome is shown in Supplementary Fig. 3A. **d** Distribution of BUSCO genes in the six *H. bakeri* chromosomes in the nxHelBake1.1 reference genomes coloured by their allocation to the seven ancestral Nigon elements defined by Gonzalez de la Rosa et al. [42]. Source data this figure can be found in the Source Data file and the GitHub repository.

(Supplementary Fig. 4). The reference genomes have high base-level accuracy, with consensus quality value (QV) scores of 55.3 (an average of one error every 389 kb) and 53.9 (an average of one error every 245 kb), and BUSCO completeness scores of 92.0 and 93.7% completeness (Table 1). Although our data show relatively little bias across GC values (Supplementary Fig. 5), we note that the completeness score of our *H. bakeri* reference genome is slightly lower than the previous *H. bakeri* reference genome[43] (94.3%), which may be due to coverage dropouts associated with long-range PCR amplification.

We also isolated and amplified polyadenylated mRNAs from two individuals (nxHelBake1 and ngHelPoly2) to facilitate protein-coding gene prediction. We sequenced the resulting cDNA libraries using PacBio Sequel IIe platform and generated 3.6 and 0.7 Gb of PacBio HiFi data per individual, with read length N50s of 1.9 kb and 1.3 kb (Supplementary Table 2). We assembled the long cDNA reads into 124,254 and 45,333 full-length transcripts that had BUSCO completeness scores of 77.7% and 49.5% (Supplementary Table 2). Prior to predicting protein-coding genes, we performed de novo repeat identification on

both genomes using EarlGrey[44]. We manually curated over 1500 predicted models for each genome using several rounds of edge extension, multiple sequence alignment, structural domain identification, and clustering (see Methods). This reduced the redundancy of the repeat models and improved their quality and classification. The final repeat libraries contained 1176 consensus sequences for *H. bakeri* and 1240 for *H. polygyrus*, allowing us to mask 64.7% and 64.0% of the genomes, respectively. After masking repetitive elements, we used a combination of evidence from short-read RNA-seq derived from pools of individuals, full-length transcripts from long-read cDNA data, and protein homology to predict 19,117 and 20,622 protein-coding genes in the *H. bakeri* and *H. polygyrus* reference genomes (Table 1). These final gene sets have 92.8% and 94.0% BUSCO completeness scores (Table 1). The inclusion of transcripts from long-read cDNA data into our final gene sets led to a substantial increase in the number of isoforms predicted per gene and in the number of genes with predicted UTRs relative to those generated using only short-read RNA-seq and/or protein homology as evidence (Supplementary Table 2). It also led to

**Table 1 | Metrics of the Heligmosomoides bakeri and Heligmosomoides polygyrus reference genomes**

| Accession information | | | | |
|---|---|---|---|---|
| Species | *H. bakeri* | *H. polygyrus* | *H. bakeri* | *H. bakeri* |
| Accession | PRJEB57615 | PRJEB57641 | PRJEB15396 | PRJEB1203 |
| Version | nxHelBake1.1 | ngHelPoly1.1 | WBPS17 | WBPS17 |
| Reference | This work | This work | Chow et al. [43] | International Helminth Genomes Consortium[66] |
| **Genome assembly metrics** | | | | |
| Span (Mb) | 649.2 | 656.6 | 697.0 | 560.7 |
| Number of scaffolds | 321 | 1,278 | 23,647 | 44,726 |
| Scaffold N50 (Mb) | 110.8 | 107.4 | 0.2 | 0.04 |
| Percentage of assembly in six chromosomes (%) | 98.6 | 96.2 | - | - |
| Number of contigs[a] | 3719 | 4738 | 56,301 | 81,726 |
| Contig N50 (Kb)[a] | 293.0 | 291.5 | 42.6 | 12.8 |
| Number of gaps | 3398 | 3460 | 32,791 | 37,001 |
| Span of Ns (kb) | 679.4 (0.1%) | 691.6 (0.1%) | 4471.1 (0.6%) | 3792.4 (0.6%) |
| QV | 55.3 | 53.9 | - | - |
| Genome BUSCO completeness (%)[b] | 92.0 | 93.7 | 94.3 | 77.0 |
| Genome BUSCO duplication (%)[b] | 1.2 | 2.5 | 1.2 | 1.1 |
| **Protein-coding gene prediction metrics** | | | | |
| Number of protein-coding genes | 19,117 | 20,622 | 23,471 | 27,459 |
| Number of transcripts | 28,195 | 24,144 | 25,215 | 27,459 |
| Gene set BUSCO completeness (%)[b] | 92.8 | 94.0 | 90.5 | 62.2 |

[a]Contig values were calculated by splitting scaffolds at ≥10 consecutive Ns.
[b]Genome and gene set completeness was assessed using BUSCO (version 5.2.2) with the nematoda_odb10 dataset (using the Augustus option when assessing genome completeness).

an increase in the number of genes with single-copy orthologues in *H. contortus* and the proportion of proteins that were within 10% of the length of the orthologous *H. contortus* protein sequence (Supplementary Table 2). We note that 1505 more genes are predicted for *H. polygyrus* than for *H. bakeri*, but this difference is likely due to technical reasons rather than biological differences. Relative to *H. bakeri*, the *H. polygyrus* genome has both higher BUSCO completeness (93.7% vs 92.0%) and duplication (2.5% vs 1.2%) scores and the gene set has substantially more single exon genes (3277 vs 2406) (Supplementary Table 2).

### The *H. bakeri* and *H. polygyrus* genomes are highly divergent

*H. bakeri* and *H. polygyrus* can be distinguished by the divergence in their ribosomal RNA cistron internal transcribed spacer 2 (ITS2) and mitochondrial cytochrome c oxidase I (COI) sequences[37]. We extracted ITS2 and COI sequences from the nuclear and mitochondrial assemblies of each individual and explored their relationship to previously generated data. As expected, we recovered our three *H. bakeri* individuals in a clade with *H. bakeri* individuals from laboratory mouse colonies in both ITS2 and COI trees (Fig. 2a; Supplementary Fig. 6). In contrast, we recovered both our *H. polygyrus* individuals in a clade alongside *H. polygyrus* individuals collected from wild *A. sylvaticus* populations in the United Kingdom and Portugal. Both ITS2 and COI showed substantial divergence between the two species, with mean identities of 96.6% and 91.1%, respectively. Our results are consistent with those of Cable et al. [37] and confirm that the individuals we sequenced are representative of laboratory and wild populations of *H. bakeri* and *H. polygyrus*.

To quantify genome-wide divergence between *H. bakeri* and *H. polygyrus*, we aligned our chromosome-level reference genomes using minimap2[45]. Only 365 Mb (56%) of the *H. bakeri* genome was covered by *H. polygyrus* alignments, which had a mean identity of 95.0% (Fig. 2b). The divergence between aligned sequences was generally lower in the middle of each chromosome except for the X (Fig. 2c). Importantly, although minimap2 was able to align sequences that showed up to 19.7% nucleotide divergence, 284 Mb (44%) of

the *H. bakeri* genome was not covered by an alignment, suggesting that many regions are too divergent to allow alignment at the nucleotide level. Despite the high level of nucleotide divergence, gene order is highly conserved between the two species (Fig. 2b), with 98.4% of neighbouring gene pairs in *H. bakeri* having colinear *H. polygyrus* orthologues. While we observe an apparent inversion on the X chromosome, the central region of the X is highly repetitive in both genomes (Supplementary Fig. 7) and the order of the contigs within this region in our reference genomes is uncertain. We estimated the average synonymous site divergence ($d_S$) between *H. bakeri* and *H. polygyrus* to be 6.6% (Fig. 2d). To place this divergence into context, we calculated the $d_S$ between two filarial nematode sister species pairs and two free-living *Caenorhabditis* sister species pairs. The divergence between *H. bakeri* and *H. polygyrus* is ~1.8 times greater than the divergence between the filarial parasites *Brugia malayi* and *Brugia pahangi* ($d_S$ of 3.6%) and ~4.5 times greater than the divergence between *Onchocerca volvulus* and *Onchocerca ochengi* ($d_S$ of 1.5%) (Supplementary Fig. 8). In contrast, both *Caenorhabditis* sister species pairs are substantially more diverged than *H. bakeri* and *H. polygyrus*: *C. briggsae* and *C. nigoni* show an average $d_S$ of 18.8% (~2.8 times greater) and *C. remanei* and *C. latens* show an average $d_S$ of 14.3% (~2.2 times greater) (Supplementary Fig. 8). We note, however, that *Caenorhabditis* species have a substantially shorter generation time than *Heligmosomoides* species (~3 days compared with ~16 days[46,47]). Using a molecular clock, we estimate that *H. bakeri* and *H. polygyrus* last shared a common ancestor ~6.4 million generations ago (see Methods), which would correspond to between 1.1 million years (assuming an average of six generations per year) and 6.4 million years (assuming an average of one generation per year). We thus regard *H. polygyrus* and *H. bakeri* as distinct species rather than demes of a single species.

### Heterozygosity is concentrated in distinct regions of the *H. bakeri* genome

The *H. bakeri* line used in laboratories throughout the world was founded from a small number of individuals and has been maintained

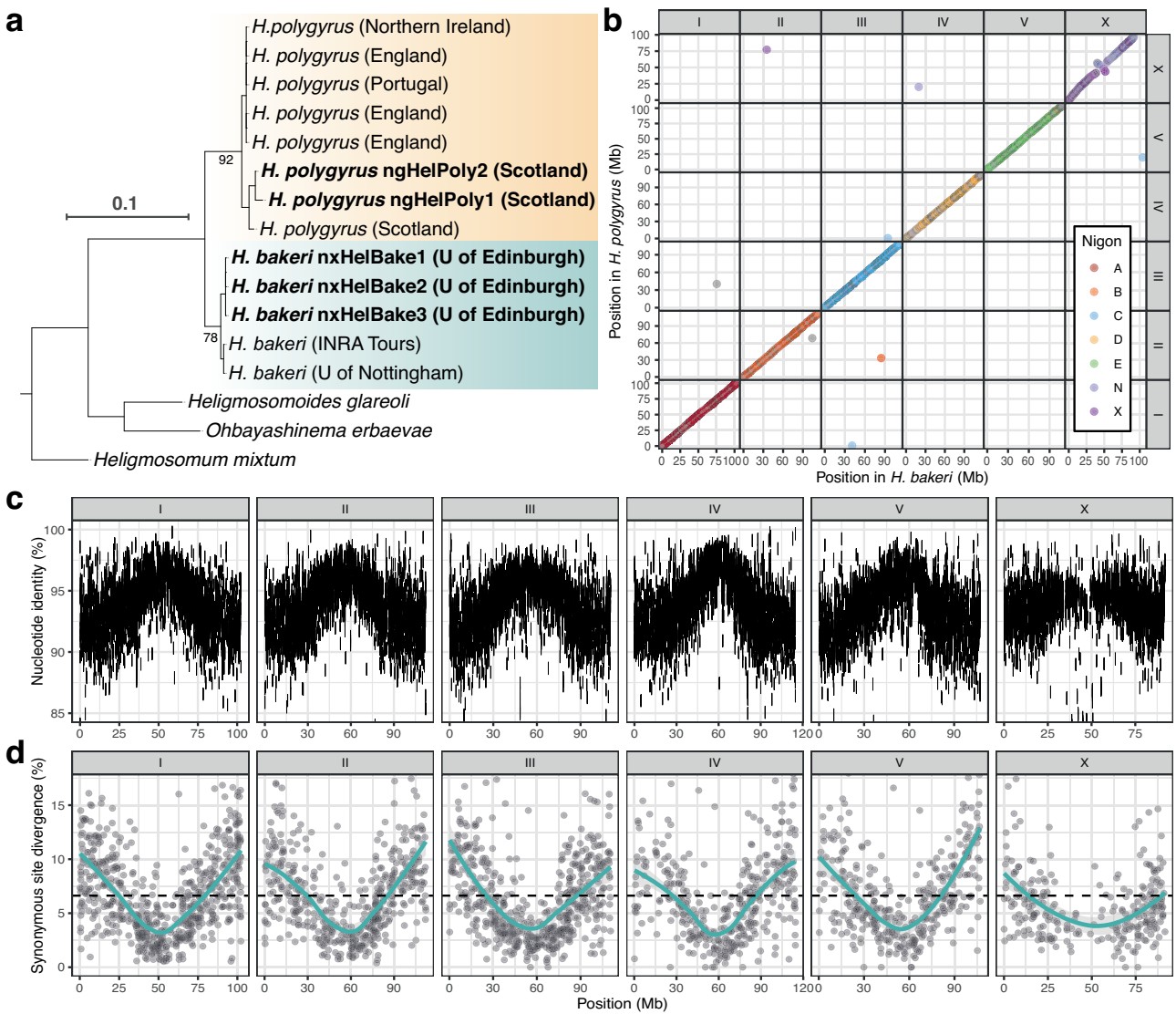

**Fig. 2 | The *H. bakeri* and *H. polygyrus* genomes are highly divergent. a** Gene tree of the ribosomal internal transcribed spacer 2 (ITS2) in laboratory isolates of *H. bakeri*, wild isolates of *H. polygyrus*, and outgroup taxa. The sequences are from Cable et al. [37] or the nuclear genomes of individuals sequenced as part of this work (highlighted in bold). The location of the laboratory or wild mouse colony from which each individual was collected is indicated. Bootstrap support values are shown for the branches subtending the *H. bakeri* and *H. polygyrus* clades. Branch lengths represent the number of substitutions per site; scale is shown. **b** The relative position of 2667 BUSCO genes in the *H. bakeri* and *H. polygyrus* genomes.

Points are coloured by Nigon element. **c** Whole-genome alignment between *H. bakeri* and *H. polygyrus* from minimap2. The position of each alignment in the *H. bakeri* reference and the nucleotide identity is shown. Only alignments that are ≥5 kb are shown. **d** Synonymous site divergence in 2670 BUSCO genes in the *H. bakeri* and *H. polygyrus* genomes. Solid represent LOESS smoothing functions fitted to the data; standard error is shown using grey shading. Dotted line indicates mean synonymous site divergence (6.6%). Source data this figure can be found in the Source Data file and the GitHub repository.

in culture since the 1950s[33]. It is therefore expected to be highly inbred. However, *k*-mer spectral analyses showed that, while the genomes of the *H. bakeri* individuals were substantially less heterozygous than those of the wild *H. polygyrus* individuals, they were not completely homozygous (Supplementary Fig. 2). To compare genome-wide patterns of heterozygosity between the two species, we called SNPs in each individual relative to the corresponding chromosome-level reference genomes. In *H. polygyrus* ngHelPoly1, heterozygous SNPs occurred on average 1 in every 155 bp (density of 0.0064) and were distributed unevenly across the genome (Fig. 3a). Heterozygosity was generally lower in the centre of each chromosome, which may be due to lower rates of recombination in these regions compared to the chromosomal arms, as observed in other nematode species[48–51]. We also identified several large homozygous regions, including three regions on chromosome I that each spanned more than 3 Mb (Fig. 3a)

that were absent in ngHelPoly2 (Supplementary Fig. 9A), which are presumably the result of random recent inbreeding within the local population. Aside from the runs of homozygosity in ngHelPoly1, the distribution and overall density of heterozygosity was similar in ngHelPoly2 (average of 1 heterozygous SNP in 145 bp; density of 0.0069) (Supplementary Fig. 9A).

We found lower levels of heterozygosity in *H. bakeri* nxHelBake1, with an average of 1 SNP every 543 bp (density of 0.0018) (Fig. 3b). As expected for a highly inbred line, the majority of the nxHelBake1 genome (which was male and therefore had a hemizygous X chromosome) was homozygous, with 70% of the autosomal genome being covered by 10 kb bins that contained no heterozygous SNPs (Fig. 3b; Supplementary Table 3). Heterozygosity was similarly low in nxHel-Bake2 and nxHelBake3, with averages of 1 SNP every 393 bp (density of 0.0025) and 1 SNP every 643 bp (density of 0.0016), respectively

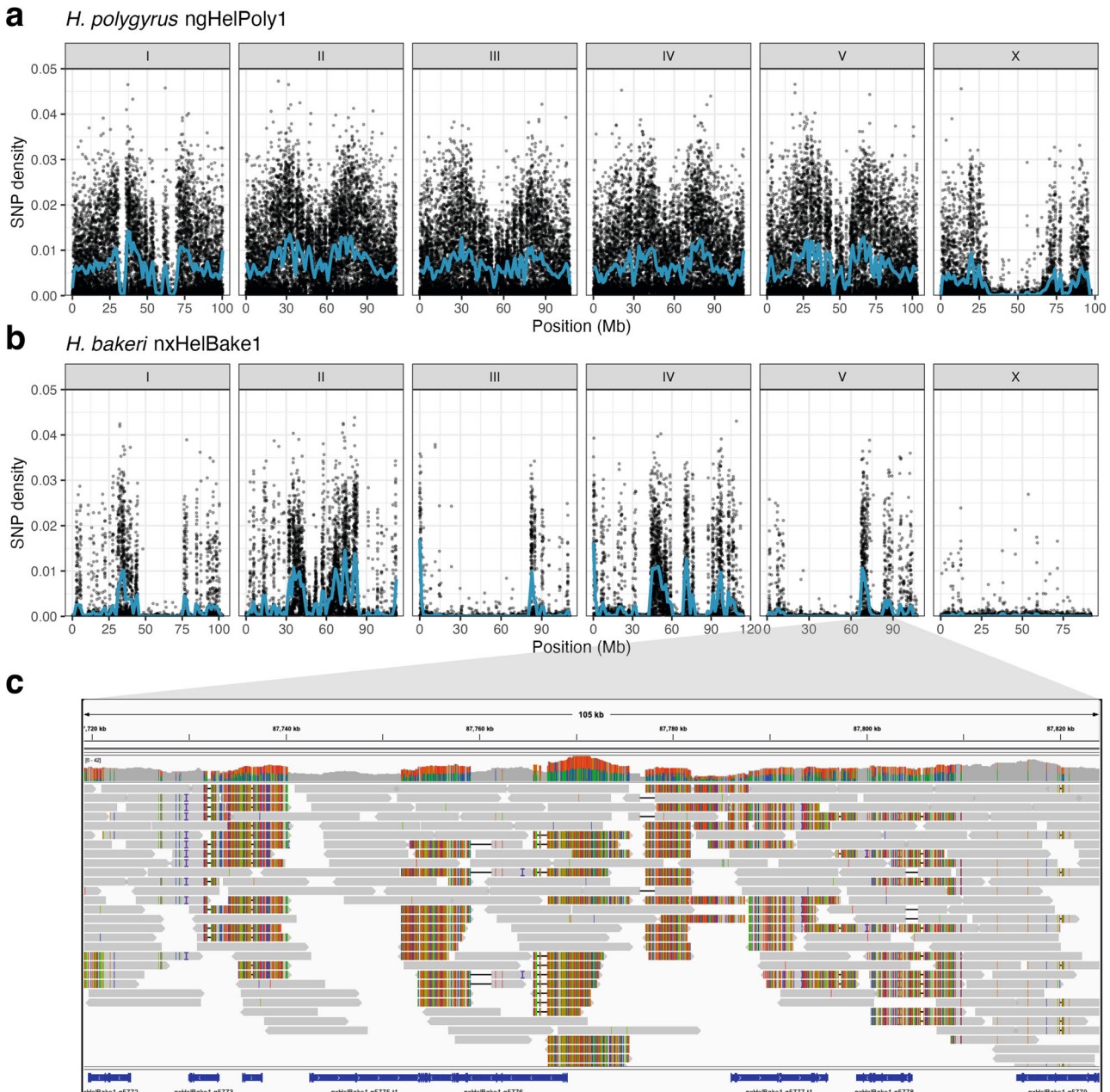

**Fig. 3 | Hyper-divergent haplotypes are widespread in the *H. bakeri* genome.** Distribution of heterozygosity in the genomes of (**a**) *H. polygyrus* ngHelPoly1 and (**b**) *H. bakeri* nxHelBake1. Points represent the density of biallelic SNPs in 10 kb windows. nxHelBake1 was male and therefore had a hemizygous X chromosome; the SNP density peaks on the X chromosome are therefore erroneous and are derived from mismapped PacBio HiFi reads. ngHelPoly1 was female. SNPs called in repeat-containing regions were filtered out and SNP density was calculated as the number of non-repetitive SNPs per non-repetitive base. Blue lines represent LOESS smoothing functions fitted to the data. **c** *H. bakeri* nxHelBake1 PacBio HiFi read alignments to the nxHelBake1.1 reference genome in a 143 kb region on chromosome V (87.7 − 87.9 Mb) that contains a 56 kb hyper-divergent haplotype. The top panel shows the coverage and the bottom panel shows aligned PacBio HiFi reads. The coloured vertical lines indicate mismatched bases at that position. An alignment between the two assembled haplotypes in this region is shown in Supplementary Fig. 10. Source data this figure can be found in the GitHub repository.

(Supplementary Fig. 9B, C; Supplementary Table 3). As in nxHelBake1, the majority of the genomes of nxHelBake2 and nxHelBake3 were homozygous, with 64% and 70% of the autosomal genomes being covered by 10 kb bins containing no heterozygous SNPs. However, we also observed many distinct regions with unexpectedly high levels of heterozygosity in all three individuals, with several regions showing SNP densities of 0.04 (1 SNP in every 25 bp) or higher (Fig. 3b; Supplementary Fig. 9B, C). Importantly, these highly heterozygous regions often occurred at different regions in the three individuals (Fig. 3b; Supplementary Fig. 9B, C) and, across all three individuals, only 44.7%

of the *H. bakeri* autosomal genome was covered by 10 kb bins that contained no heterozygous SNPs.

## Hyper-divergent haplotypes are widespread in the *H. bakeri* genome

Recent work has shown that the genomes of selfing *Caenorhabditis* nematodes are punctuated with hyper-divergent haplotypes[16–19]. To determine whether the heterozygous regions in the *H. bakeri* genome contained hyper-divergent haplotypes, we manually inspected PacBio HiFi read alignments within heterozygous regions and found many

where reads from the alternate haplotype aligned with high numbers of variants or failed to align at all. For example, in one highly heterozygous region on the right arm of chromosome V (V:87.7–87.8 Mb), we found large gaps in read alignments from the alternate haplotype that had corresponding drops in read coverage and that contained no heterozygous variants (Fig. 3c). Consistent with this, alignments between the two assembled haplotypes in this region showed many regions where nucleotide identity was less than 90% (Supplementary Fig. 10). To quantify the genome-wide distribution of hyper-divergent haplotypes in the *H. bakeri* genome, we developed a pipeline to identify highly divergent regions in alignments between each haplotype assembly and the nxHelBake1.1 reference genome and optimised our approach using the previously defined *C. elegans* hyper-divergent haplotype coordinates[17] (Supplementary Fig. 11; see Methods). In nxHelBake1, our approach identified 683 distinct regions that contained a hyper-divergent haplotype, which collectively spanned 3.6% (23.4 Mb) of the autosomal genome (Supplementary Fig. 12; Supplementary Table 4). These regions contained 665 protein-coding genes (3.5% of all genes) and contained 36.6% of the heterozygous SNPs called in nxHelBake1. The largest divergent haplotype was found on the right arm of chromosome IV and spanned 391 kb (IV:102.8–103.2 Mb). In the nxHelBake2 and nxHelBake3 primary and alternate assemblies, we classified between 167 and 697 hyper-divergent haplotypes, spanning between 4.5 and 27.4 Mb (0.7–4.2% of the reference genome) (Supplementary Fig. 12; Supplementary Table 4). Across all three individuals, we identified 1703 non-overlapping regions that contained a hyper-divergent haplotype (Supplementary Table 4), which collectively spanned 62.0 Mb (9.6% of the genome) and contained 1734 genes (9.1% of all genes).

In *C. elegans*, hyper-divergent haplotypes are enriched in genes associated with environmental response, including sensory perception and response to pathogens and xenobiotic stress[17]. We performed gene ontology (GO) term enrichment on the 1734 genes present in the hyper-divergent regions across the three *H. bakeri* individuals and found a total of 20 significantly enriched GO terms (*p*-value < 0.05), several of which have previously been associated with parasite functions (Supplementary Fig. 13; Supplementary Table 5). The most significantly enriched GO term across all three categories was 'cell surface' (GO:0009986, *p*-value = 0.000076). Fifteen proteins from the hyper-divergent regions were associated with this GO term (24.5% of all proteins associated with this GO term in the genome) and all belong to the transthyretin-related (TTR) protein family. TTRs are abundant in the ES products of several strongyle nematodes, including *H. bakeri*[52] and some have been shown to modulate the host immune response by binding host cytokines[53]. In addition to TTRs, two terms associated with protease inhibitors were significantly enriched (GO:0008191, 'metalloendopeptidase inhibitor activity', *p*-value = 0.0094 and GO:0030414, 'peptidase inhibitor activity', *p*-value = 0.0363) and the proteins associated with these terms included tissue inhibitors of metalloproteinases (TIMP) and Kunitz/Bovine pancreatic trypsin inhibitors. Protease inhibitors are hypothesised to provide protection from degradation by host proteases and to manipulate the host immune response[54]. In the canine hookworm *Ancylostoma caninum*, a TIMP protein is the most abundant protein in the excreted/secreted (ES) products and inhibits host proteases[55,56]. Seven distinct Kunitz inhibitors have been detected in the ES products of *H. bakeri*[52] and a secreted Kuntiz inhibitor from the human hookworm *Ancylostoma ceylanicum* has been shown to inhibit a broad range of host proteases[57,58]. The term 'superoxide metabolic process' was also significantly enriched in proteins from hyper-divergent haplotypes (GO:0006801; *p*-value = 0.0108) and all three proteins associated with this term are copper/zinc superoxide dismutates (Cu/Zn SODs). Cu/Zn SODs are present in the ES products of several parasitic nematodes and are believed to protect parasitic nematodes from reactive oxygen species generated by the host immune response[59]. In addition to the

terms associated with host-parasite interaction protein families, we also found that the term 'chloride channel' was significantly enriched (GO:0005254, *p*-value = 0.0054) and is associated with seven proteins found in hyper-divergent regions, six of which belong to the bestrophin family of membrane-bound calcium-activated chloride channels[60]. Several bestrophins are expressed in the *C. elegans* nervous system[60] and a bestrophin homologue was upregulated in ivermectin-resistant populations of *H. contortus*[61]. As in *C. elegans*, the term 'sensory perception of chemical stimulus', associated with G-protein coupled receptors, is significantly enriched in proteins from hyper-divergent regions (GO:0007606, *p*-value = 0.0232).

## Ancient genetic diversity in genes associated with host-parasite interactions

Although the origin of the hyper-divergent haplotypes in *C. elegans* is unknown, they are hypothesised to be the product of balancing selection acting over long periods of evolutionary time[16,17]. Importantly, the divergence between hyper-divergent *C. elegans* alleles was shown to be comparable to that between closely related species[17]. We therefore hypothesised that some of the hyper-divergent haplotypes in the *H. bakeri* genome might predate the split between *H. bakeri* and *H. polygyrus*. To test this hypothesis, we generated a draft genome assembly for the outgroup species *Heligmosomum mixtum*, isolated from bank voles (*Myodes glareolus*), using short read data and generated gene trees for each hyper-divergent gene in *H. bakeri* and its orthologs in *H. polygyrus* and *H. mixtum*. We identified 569 genes (49.6% of the 1147 genes found in hyper-divergent haplotypes in *H. bakeri* for which we could predict proteins in at least one *H. polygyrus* haplotype and in *H. mixtum*) where the *H. bakeri* and *H. polygyrus* sequences did not form separate clades (Supplementary Data 1; see Methods). As expected, genes that showed evidence of haplotype sharing were clustered in the genome, with 42.4% being direct neighbours. Within this set of genes, we found many genes with known roles in host-parasite interactions (Supplementary Data 1).

For example, in a ~287 kb region on chromosome I in the *H. bakeri* genome (I:8.92-9.19 Mb), we identified two neighbouring genes that showed evidence of haplotype sharing. Within this region, two genes (nxHelBake1.g10196 and nxHelBake1.g10198) are members of the *Ancylostoma*-secreted protein family (ASPs), which are highly abundant in ES products of *H. bakeri* and other parasitic nematodes[52,62–64]. We manually inspected the read alignments in this region and found that nxHelBake1 was homozygous for the reference haplotype, whereas nxHelBake2 and nxHelBake3 were heterozygous or homozygous for an alternate haplotype that is hyper-divergent from the reference (Supplementary Fig. 14). Reads from the alternate haplotype do not align to nxHelBake1.g10196 and align only partially to nxHelBake1.g10198 (Supplementary Fig. 14). By identifying the contigs that corresponded to the hyper-divergent haplotype in the nxHelBake2 primary assembly, we found that the hyper-divergent haplotype is nearly 100 kb larger than the reference haplotype with the alignment suggesting the presence of a divergent inverted duplication (Fig. 4a). Within the alternate haplotype, we found two genes with homology to nxHelBake1.g10195 (which contains a growth factor receptor domain) and two genes with homology to the ASP nxHelBake1.g10196 (Fig. 4b), all of which were supported by RNA-seq reads derived from pools of individuals (Supplementary Fig. 15). The ASP nxHelBake1.g10196 and its homologues in the alternate haplotype have highly divergent protein sequences (69–75% identity). Although we did not find copy number variation in the other ASP gene in this region (nxHelBake1.g10198), the two hyper-divergent alleles share only 78% amino acid identity. In read alignments to the orthologous region in the *H. polygyrus* genome, we find similar levels of divergence to that within *H. bakeri*, including alignment gaps that cover the *H. polygyrus* orthologue of the ASP nxHelBake1.g10196 (Supplementary Fig. 16). In pairwise alignments of the five assembled haplotypes from both species, the

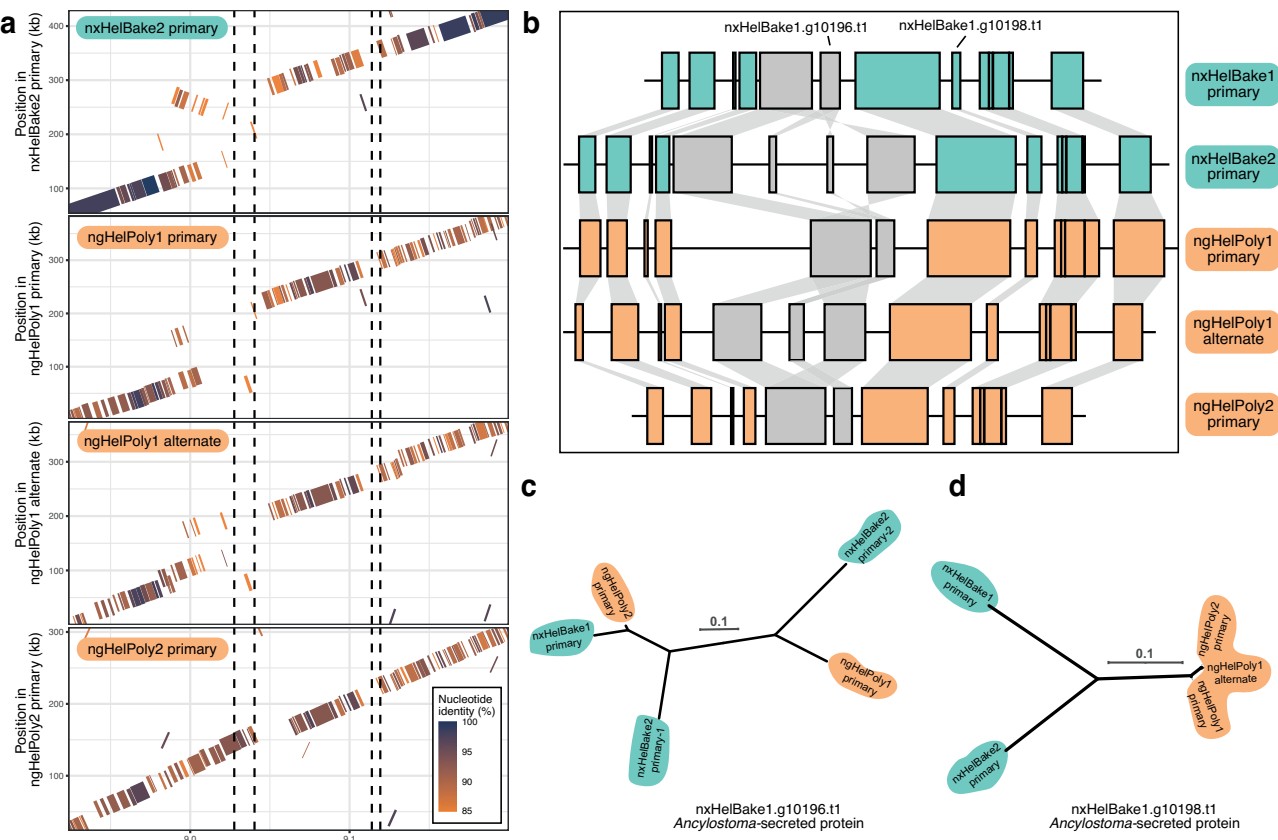

**Fig. 4 | Ancient genetic diversity in two members of the *Ancylostoma*-secreted protein family.** A 287 kb region on chromosome I of *H. bakeri* (I:8.92–9.20 Mb) containing two ASP loci that show evidence of haplotype sharing. **a** Nucleotide alignments between the alternate *H. bakeri* haplotype (represented by nxHelBake2 primary), each *H. polygyrus* haplotype (ngHelPoly1 primary, ngHelPoly1 alternate, ngHelPoly2 primary), and the nxHelBake1 reference haplotype. Alignments are coloured by their nucleotide identity. Repetitive alignments are not shown. The position of the two ASP loci (nxHelBake1.g10196 and nxHelBake1.g10198) are indicated by dotted lines. **b** Gene contents of each haplotype. The alternate *H. bakeri* haplotype (represented by nxHelBake2 primary) has two homologues of

nxHelBake1.g10195 and nxHelBake1.g10196, consistent with the divergent inverted duplication in (**a**). Green (*H. bakeri*) and orange (*H. polygyrus*) boxes represent genes that have a 1:1 relationship across all haplotypes; grey boxes represent genes that have non 1:1 relationship (nxHelBake1.g10195 and nxHelBake1.g10196). Links between genes represent homologous relationships. Gene trees of (**c**) nxHelBake1.g10196.t1 and its homologues and (**d**) nxHelBake1.g10198.t1 and its homologues. Gene trees were inferred using IQ-TREE under the LG + Γ substitution model. Scale is shown in substitutions per site. Outgroup not shown. Source data this figure can be found in the GitHub repository.

nxHelBake1 primary and ngHelPoly2 primary haplotypes were the only pair that showed alignment in the region containing nxHelBake1.g10196. Consistent with this, the gene tree of nxHelBake1.g10196 and its homologues recovered nxHelBake1 primary and the ngHelPoly2 primary sequences as more closely related to each other than to any other homologues. The two protein sequences share 91% identity, 20% higher than that between the two hyper-divergent *H. polygyrus* alleles (Fig. 4c).

In another region on chromosome III in *H. bakeri* (III:0.58-1.15 Mb) that spans ~570 kb, we identified ten genes with evidence of haplotype sharing (Supplementary Data 1). Four genes in this region are members of the novel secreted protein (NSP) family that is abundant in *H. bakeri* ES products[52]. Read alignments of the two *H. polygyrus* individuals to this region of the *H. bakeri* reference genome show substantially lower levels of divergence than those of nxHelBake2 and nxHelBake3 (Supplementary Fig. 17A). Consistent with this, the assembled reference *H. bakeri* haplotype and reference *H. polygyrus* haplotype share higher nucleotide identity than the two hyper-divergent *H. bakeri* haplotypes (Supplementary Fig. 17B) and gene trees of the NSP proteins recover the *H. bakeri* reference haplotype as more closely related to all *H. polygyrus* haplotypes than to any other *H. bakeri* haplotype (Supplementary Fig. 17C, D). In another region on chromosome II (20.9–21.1 Mb) that contains six TTRs, we identified three genes as

showing evidence of haplotype sharing (Supplementary Fig. 18C–E; Supplementary Data 1), which is consistent with read and assembled haplotype alignments (Supplementary Fig. 18A, B). The two hyper-divergent *H. bakeri* nxHelBake1.g653.t1 alleles share 95.7% amino acid identity, whereas the reference *H. bakeri* and alternate *H. polygyrus* alleles share 97.1% identity. We also found evidence for haplotype sharing within a homolog of the aminopeptidase H11, an antigen that is used in a commercially available vaccine against the sheep parasite *H. contortus*[65] (Supplementary Fig. 19).

The haplotype sharing we observed between *H. bakeri* and *H. polygyrus* could result from maintenance since the last common ancestor by long-term balancing selection, leading to trans-specific polymorphism, or recent gene flow between the two species. If the hyper-divergent haplotypes have been maintained since the last common ancestor, the divergence between the shared haplotypes would be expected to be similar to the genome-wide average, whereas haplotypes that are the result of recent gene flow would be expected to show less divergence than the genome-wide average. To distinguish between these two scenarios, we identified one-to-one orthologues in haplotypes that were shared between nxHelBake1 primary and ngHelPoly1 primary and compared their synonymous site divergence ($d_S$) to the divergence between the 9,753 one-to-one orthologues found in non-divergent regions of the genome. We find similar levels of

 

divergence in one-to-one orthologues within shared haplotypes (mean $d_S = 0.069$, $N = 189$) to the average genome-wide divergence (mean $d_S = 0.074$, $N = 9,753$). Importantly, the $d_S$ in shared haplotypes is not significantly different from the genome-wide average (two-sided Wilcoxon; $p$-value = 0.1362) (Supplementary Fig. 20). Therefore, the divergence within shared haplotypes is consistent with maintenance since the last common ancestor of *H. bakeri* and *H. polygyrus*.

## Discussion

### New approaches for parasite genomics

Generating reference genomes for parasitic nematodes has been hindered by the difficulty of obtaining material for sequencing, the small size of individual parasites, and the high levels of genetic diversity circulating in nematode populations. Here, we employed the recently developed PiMmS protocol to sequence the genomes of several individuals of *H. bakeri* and *H. polygyrus* using PacBio HiFi long-read technology. Where we obtained sufficient coverage (≥25-fold), our single nematode assemblies are highly complete and more contiguous than previous versions of the *H. bakeri* reference genome[43,66]. Importantly, our use of highly accurate long-read technology allowed us to generate haplotype-resolved assemblies, enabling the study of hyperdivergent haplotypes. We also sequenced Hi-C libraries derived from pools of individuals, which allowed us to order and orient the primary assemblies from single nematodes into complete chromosomes. However, we note that our single nematode assemblies are not as contiguous as other nematode genomes that have been sequenced with long-read technology[19,20,42], many of which have contig N50s of a megabase or longer. Although this may be in part due to the large and repetitive nature of the *Heligmosomoides* genomes we sequenced, we believe that the primary reason is our use of long-range PCR amplification. When used to amplify whole genomes, PCR is known to introduce coverage biases across the genome[67], which can lead to coverage dropouts and therefore assembly gaps. Although PiMmS shows relatively little GC coverage bias, the presence of regions that are recalcitrant to amplification by PCR could explain why our chromosome-level reference genomes are marginally less biologically complete (92.0% and 93.7% versus 94.3%) than the previous version of the *H. bakeri* reference genome, despite being more contiguous.

However, even with these drawbacks, PiMmS and protocols such as the PacBio ultra-low input protocol, which has been used to sequence single arthropods and nematodes[29,68,69], represent an exciting new opportunity for parasite genomics. By sequencing individual parasites, we can circumvent the issues that arise when trying to assemble genome datasets derived from pools of genetically polymorphic individuals[20,24]. This is particularly valuable for parasites from wild animals, as in the case of *H. polygyrus*, which are expected to contain high levels of genetic diversity and for which obtaining large amounts of material for sequencing may be more challenging. Importantly, the PiMmS protocol has been used to sequence nematodes as small as *C. elegans* (Laumer CE, pers. comm.; an adult *C. elegans* hermaphrodite is 1 mm long and contains ~200 pg of DNA), meaning this protocol will likely be applicable to even the smallest parasitic nematodes. Although we targeted adults for sequencing (which we obtained via dissection post-mortem), it may be possible to use PiMmS to sequence the genomes of infective juveniles isolated from faeces, which would enable the sequencing of parasites without the need to dissect a host. These new technologies, combined with large-scale efforts to sequence the Earth's biodiversity[22,23], mean that we will likely see a rapid increase in the availability and quality of genomic datasets for parasitic nematodes in the coming years.

### Genomic evidence supports the species status of *H. bakeri* and *H. polygyrus*

Despite commonly being referred to as *H. polygyrus* in the literature, several lines of evidence have been presented that suggest that the laboratory model organism is distinct from the parasites found in European wood mice, including morphological differences[35,70], differences in host specificity[36], and substantial divergence in nuclear and mitochondrial markers[37,71–73]. Here, we have sequenced the genomes of *H. bakeri* individuals collected from the research strain from a laboratory mouse population at the University of Edinburgh and *H. polygyrus* individuals from a wild *A. sylvaticus* population in Scotland and have shown that their genomes are highly divergent. In our whole-genome alignments between the two species, over half of the *H. bakeri* genome is not covered by *H. polygyrus* alignments. The divergence between *H. bakeri* and *H. polygyrus* is 1.8 and 4.5 times greater than the divergence between *B. malayi* and *B. pahangi* and between *O. volvulus* and *O. ochengi*, respectively. Importantly, the ITS2 and COI sequences derived from our reference genomes associate coherently within two well-supported clades comprising *H. bakeri* individuals from other laboratories and *H. polygyrus* individuals from multiple populations in Europe[37]. Using a molecular clock, we estimate that *H. bakeri* and *H. polygyrus* last shared a common ancestor at least 1 million years ago. We therefore believe that there is now overwhelming evidence that *H. bakeri* and *H. polygyrus* are separate species and we encourage the use of the name *H. bakeri* to refer to the laboratory model nematode strain, as previously suggested[74,75].

The existence of two closely related parasitic nematodes, one of which is a well-established model organism, provides a promising opportunity to understand the evolution and genetic basis of gastrointestinal nematode infection. Of particular interest is in understanding the molecular basis of how *H. bakeri* is able to establish infections in both *M. musculus* and *A. sylvaticus*, while *H. polygyrus* can only establish infections in *M. musculus* if the hosts are immunosuppressed[36]. Moreover, a resolution to this taxonomic debate would also allow for a renewed focus on the true natural history of *H. bakeri*. The collection of additional *H. bakeri* isolates from wild populations could provide a valuable source of natural phenotypic and genetic variation, including in parasite-relevant traits, that could form the basis of quantitative genetic studies in *H. bakeri*, an approach that has proved valuable in *C. elegans* and *H. contortus*[76,77].

### Hyper-divergent haplotypes and their role in parasite biology

Recent work in free-living *Caenorhabditis* nematodes found that their genomes contain hundreds of hyper-divergent haplotypes that are enriched in environmental response genes[16–19]. Although these regions are believed to be the product of long-term balancing selection, their exact origins are unknown[16,17]. Moreover, whether these regions are common features of nematode genomes, including in those of parasitic species, is unknown. Here, we have discovered that the *H. bakeri* genome contains hundreds of hyper-divergent haplotypes despite over 70 years in the laboratory. These haplotypes contain 9.1% of the protein-coding genes predicted in the *H. bakeri* genome and are enriched in several gene families that have previously been associated with parasite biology, including protease inhibitors, TTRs, and Cu/Zn SODs. By studying a subset of these regions in detail, we found extensive protein sequence divergence between hyper-divergent alleles and copy number variation in proteins that are known to be highly abundant in the ES products of *H. bakeri* and other nematodes, including ASPs and NSPs[52,62–64]. We also found that a substantial proportion of these haplotypes are shared between *H. bakeri* and *H. polygyrus* and show levels of divergence that are consistent with long-term maintenance by balancing selection.

What might cause balancing selection to act over such long periods of evolutionary time? The selection pressures exerted by the host immune response offer a potential explanation. Parasite antigens that lead to acquired immunity are expected to evolve under negative frequency-dependent selection, whereby rare alleles (i.e., those where the level of acquired immunity in host populations is low) have a fitness advantage over common alleles[2,3,78]. This leads to the

 

maintenance of alleles that would otherwise be lost due to genetic drift. If this selection acts over long periods of time, it can result in trans-specific polymorphism, whereby polymorphisms are shared between closely related species[4]. Trans-specific polymorphism between humans and chimpanzees has been found at pathogen response loci, including the MHC, and between *Capsella* species in immunity-related genes[5,79]. In *Heligmosomoides*, we have found evidence for trans-specific polymorphism in several regions containing genes previously shown to interact with the host immune response, including in ASPs, NSPs, and TTRs. We therefore speculate that the immune response of the host has led to stable selection for rare alleles at these regions since the last common ancestor of *H. bakeri* and *H. polygyrus*. These same selective forces may also explain highly diverse regions in other nematode genomes. For example, in *Strongyloides ratti*, a parasite of rats, genes that are up-regulated in parasitic life-stages, particularly astacin-like metallopeptidases and ASPs, are enriched in regions of high SNP density[80]. It is also possible that many of the hyper-divergent regions identified in *C. elegans* are a product of selection exerted through host-parasite interactions, albeit with *C. elegans* as the host rather than the parasite. Consistent with this, the majority of the *pals* genes, which are involved in pathogen response[81], are found in hyper-divergent haplotypes and genes that are differentially expressed upon exposure to natural *C. elegans* pathogens are strongly enriched in these regions[17]. However, although this model may explain some of the regions we have studied, it seems unlikely to explain all hyper-divergent haplotypes in *H. bakeri* and *H. polygyrus*. Although several of the enriched gene families do have relevance for parasite biology, including chloride channels and GPCRs, they are not known to directly interact with the host immune response and so should not be subject to selection imposed by Red Queen dynamics. Instead, selection pressures imposed by environmental fluctuations (both within and outside a host), interaction with other parasites and bacteria within the host gut, and/or interspecific conflict may account for some of the regions we discovered. Additionally, selfish toxin-antidote elements, which are known to be common in nematode genomes[18,82], can lead to many of the same observed features, including high levels of divergence, presence-absence variation in genes, and signatures of balancing selection[17,83]. Fortunately, *H. bakeri*'s status as a model organism offers an opportunity to experimentally determine what role these regions might play in parasite biology and if their maintenance could be explained by selection exerted by the host-parasite interaction.

Regardless of the drivers of the origin and maintenance of these regions in natural populations of *H. bakeri* and *H. polygyrus*, we are surprised by the extent of hyper-divergent haplotypes (and, more generally, heterozygous regions) in the *H. bakeri* genome after over 70 years in laboratory culture. The effective population size ($N_e$) of laboratory populations is expected to be extremely small, meaning balancing selection would have to be extremely strong to counter the effects of genetic drift. It is likely that some of the heterozygous regions in the *H. bakeri* genome may be maintained due to the presence of recessive deleterious or lethal alleles, which prevent certain regions of the genome from becoming homozygous. Attempts to inbreed free-living *Caenorhabditis* nematodes prior to genome sequencing often result in inbreeding depression and substantial proportions of their genomes have been found to remain heterozygous even after 20 generations of full-sibling mating[24,84]. Although bottlenecks and drift should mean that very little heterozygosity is retained in the laboratory strain of *H. bakeri*, it is possible that some of the observed residual heterozygosity exists purely due to chance. Future work to simulate the loss of heterozygosity over time, given the estimated laboratory population size and the number of generations since isolation, could reveal whether the distribution of heterozygosity in the *H. bakeri* genome is significantly different from neutral expectations.

The existence of these regions in the *H. bakeri* genome, and in parasite genomes in general, has several important implications. First, *H. bakeri* is a well-established model organism but our results suggest that different individuals (and therefore different experimental groups) can have substantial genetic differences, including in parasitism-relevant regions of the genome, which could influence experimental results. In addition, because *H. bakeri* is maintained live and shared between laboratories in an ad hoc way (rather than sourced from a single, cryopreserved stock, like *C. elegans*), it is possible that genetic differences exist between populations in different laboratories, which may lead to different experimental outcomes. Indeed, several studies have shown that laboratory selection can modify various traits in *H. bakeri*, including sensitivity to the host immune response and response to ivermectin[85–88]. Second, previous studies of genome-wide patterns of genetic diversity in parasitic nematodes have relied on mapping short-read data to a reference genome[14,89]. As shown here and in *C. elegans*[17], the levels of divergence between hyper-divergent haplotypes prevent accurate read alignment, meaning only long-read sequencing and de novo assembly-based approaches can successfully characterise their contents. Lastly, the presence of hyper-divergent haplotypes in parasite populations poses challenges for the development of effective anti-nematode vaccines. Anthelmintic drugs are widely used to control parasitic nematodes of both humans and livestock, but resistance is a growing problem globally[90], leading to an increased focus on alternate control measures. Efforts to develop vaccines against human hookworms (*Ancylostoma duodenale* and *Necator americanus*), which infect hundreds of millions of people per year[91], have used ASPs as vaccine antigens[92] and one of the two antigens used in a commercially available vaccine against *H. contortus* is H11[65]. We found hyper-divergent haplotypes and evidence of trans-specific polymorphisms at both ASP and H11 homologues in *Heligmosomoides*. If hyper-divergent haplotypes exist at loci targeted by vaccines, resistance is likely to develop quickly. Understanding the patterns and levels of genetic diversity present in parasite populations will therefore be a key step in the development of effective anti-nematode vaccines.

## Methods

### Ethical statements and Licence numbers

All *H. bakeri* and *H. polygyrus* protocols and procedures were approved by The University of Edinburgh's Animal Welfare and Ethical Review Board and undertaken in accordance with a UK Home Office Project License following the 'principles of laboratory animal care' (National research council., 2003). *H. bakeri* individuals were collected from adult mice under the Home Office license PPL-P635073CF. *H. polygyrus* individuals were collected from adult *A. sylvaticus* mice under the Home Office license PPL-PP4913586. The *H. mixtum* procedures were approved by the Local Ethical Committee no. 1 in Warsaw, Poland (decision 304/2012).

### Tree of Life identifiers

We use Tree of Life identifiers (ToLIDs) to refer to *H. bakeri* and *H. polygyrus* individuals that were sequenced at the Sanger Institute (see https://id.tol.sanger.ac.uk/ for more information). The ToLIDs for *H. polygyrus* were created prior to updating the nematode taxonomy, meaning they are prefixed with ng (e.g., ngHelPoly1), which represents Nematoda + Strongylida, whereas the *H. bakeri* individuals are prefixed with nx (e.g., nxHelBake1), which represents Nematoda + Chromadorea.

### Collection of adult *H. bakeri* from laboratory mice

CBA × C57BL/6 F1 (CBF1) mice were infected with 400 L3 infective-stage *H. bakeri* larvae by oral gavage and adult nematodes were collected from the small intestine 14 days post-infection. The nematodes were washed and maintained in serum-free medium in vitro[93] for 24 h

prior to harvesting. Individual nematodes were stored in cryovials at −80 °C prior to single nematode sequencing (see below). A pool of these individuals weighing ~15 mg (~380 individuals) was generated and stored at −80 °C prior to Hi-C library preparation (see below).

### Field trapping of *A. sylvaticus*

*A. sylvaticus* were trapped on ten grids over two sites in a 3-weekly rotation from May to December 2019. The two sites were based within 14 km from Edinburgh, Scotland: one in Penicuik Estate (55 48'56.5"N 3 15'23.1"W) and the other near Loanhead (55 52'09.3"N 3 08'34.4"W). Trapping occurred 3 days a week. Each site contained five 60 m × 60 m grids. Traps were baited with birdseed, carrot and cotton bedding. These were set at dusk each day and checked for animals the following morning. Each animal was weighed, sexed, measured, given a fat score (1–5), and checked for ectoparasites. All traps were kept to harvest the faecal matter.

### Collection of wild adult *H. polygyrus*

A subset of the trapped *A. sylvaticus* were sacrificed using Schedule 1 methods. The gastrointestinal tracts were removed and the small intestine was examined for adult *H. polygyrus*, which were carefully removed with forceps, washed, and maintained in serum-free medium in vitro[93]. Individual *H. polygyrus* were stored in cryovials at −80 °C until needed. Individuals collected in Loanhead were used for single nematode sequencing (see below).

### Wild *H. polygyrus* larval cultivation

Fresh faeces were collected from all the traps that contained a wood mouse and pooled. This was soaked in sterile water, homogenised and then mixed at a 1:1 ratio with inactivated charcoal to allow any eggs to hatch into larvae using the methods described previously[93]. Briefly, a thin layer of the charcoal-faeces mix was smeared onto damp Whatman filter paper and placed onto a petri-dish. The petri dishes were then stored in a humid box, at room temperature, in the dark for up to 3 weeks. Every 7 days the filter paper was carefully lifted and the petri dishes were rinsed with 5 ml sterile water and the larvae were collected into a 50 ml falcon tube. These L3 larvae were washed three times with distilled water and then stored at 4 °C until needed. A subset of these larvae was sent for IDEXX testing to ensure they were not contaminated with any wild pathogens.

### Infection of wood mice

The University of Edinburgh maintains an originally wild-derived, but now laboratory reared, outbred colony of wood mice (*A. sylvaticus*). Six co-housed wood mice aged 6–8 weeks from our colony were administered 150 L3 per 150 μl $H_2O$ via oral gavage on day 0. These animals were checked daily and had access to food and water *ad libitum*. From day 9 to day 21 post-infection, the bedding of the mice was collected daily and fresh bedding was supplied. The soiled bedding was sieved to collect all the faecal pellets and these pellets were mixed with inactivated charcoal as detailed above. On day 21 post-infection, the mice were culled via Schedule 1 methods. n. The GI tracts were dissected and any adult nematodes found were processed as above. A pool of these individuals weighing ~15 mg (~380 individuals) was generated and stored at −80 °C prior to Hi-C library preparation (see below).

### Collection of *H. mixtum* from bank voles

We collected *H. mixtum* from bank voles (*Myodes glareolus*) captured in Urwitałt, in northeast Poland (53° 47' 56.50"N, 21° 38' 44.4"E) in Autumn 2014. Rodents were live-trapped in wooden traps with grain and carrot or apple as bait. Each animal was weighed, sexed, measured, and sacrificed by isoflurane overdose and cervical dislocation. We carefully examined intestines under a stereoscopic microscope in search of nematodes of any species. We stored individual nematodes in 70% ethanol at −20 °C until DNA extraction. For the current analysis, we used two male *H. mixtum* found in different host individuals. We isolated DNA from individual nematodes using the Nucleospin Tissue kit (MN) or DNeasy Blood and Tissue DNA extraction kit (Qiagen). Edinburgh Genomics prepared Illumina Nextera libraries for two individuals (Hm2 and Hm16), which were sequenced on the Illumina HiSeq 2500 platform using 125 bp PE sequencing.

### Single nematode genome and transcriptome sequencing

We used the PiMmS protocol to extract and amplify the DNA from three individuals of *H. bakeri* (nxHelBake1, nxHelBake2, and nxHelBake3) and two individuals of *H. polygyrus* (ngHelPoly1 and ngHelPoly2). After the extraction of the DNA, each sample was purified and sheared gDNA to ~10 kb. We used between 11 and 18 PCR cycles during long-range PCR amplification. These amplified libraries were prepared using the PacBio SMRTbell Express Template Prep 2.0 kit as per the manufacturer's instructions. The final libraries were purified using 1.0X ProNex beads before elution. We carried out size selection using the BluePippin gel electrophoresis system with 0.75% DF Marker S1 6–10 kb High pass cassettes with an 8 kb size cut-off. We quantified the DNA concentration of each size-selected library using the Qubit dsDNA HS assay and assessed fragment length distribution using a FEMTO pulse. Each library was sequenced by the Scientific Operations core at the Wellcome Sanger Institute on a single PacBio Sequel IIe flow cell, except for ngHelPoly1, which was sequenced on two flow cells.

For two individuals (nxHelBake1 and ngHelPoly2), we isolated and amplified cDNA libraries alongside the gDNA libraries using the PiMmS protocol. After reverse transcription, we used 20 and 18 PCR cycles to amplify the nxHelBake1 and ngHelPoly2 cDNA libraries, respectively. The final libraries were purified and assessed using DNA concentration and fragment length as previously described. nxHelBake1 was sequenced on one-half of a PacBio Sequel IIe flow cell and ngHelPoly2 was sequenced on one-third of a PacBio Sequel IIe flow cell by the Scientific Operations core at the Wellcome Sanger Institute.

### Short-read mRNA-seq for *H. polygyrus*

Total RNA from *H. polygyrus* adult nematodes was isolated using the miRNAeasy kit (Qiagen) and visualised on a bioanalyzer (RIN > 7.0) prior to library preparation using TruSeq Stranded Total RNA Library PrepGold with ribodepletion. Libraries were sequenced using NextSeq 2000 platform for 100 cycles by Edinburgh Genomics.

### Hi-C library preparation and sequencing

Hi-C library preparation and sequencing was performed by the Scientific Operations core at the Wellcome Sanger Institute. Two ~30 mg pellets of frozen mixed-sex adult *H. bakeri* and *H. polygyrus* nematodes were processed using the Arima Hi-C version 2 kit following the manufacturer's instructions and Illumina libraries were prepared using the NEBNext Ultra II DNA Library Prep Kit for Illumina. Each library was sequenced on one-eighth of a NovaSeq S4 lane using paired-end 150 bp sequencing.

### Single nematode genome assembly

We used lima v2.6.0 (available at https://lima.how/) to remove TruSeq adapter sequences and pbmarkdup v1.0.2 (available at https://github.com/PacificBiosciences/pbmarkdup) to remove PCR duplicates from the genomic PacBio HiFi reads. We removed any remaining PacBio adapter sequences using HiFiAdapterFilt[94]. We used Jellyfish 2.3.0[95] to count *k*-mers of length 31 in each read set and GenomeScope 2.0[96] to estimate genome size and heterozygosity. To identify and remove contamination, we first generated preliminary assemblies for each individual using hifiasm v0.16.1-r375[97]. We aligned Hi-C reads to each hifiasm primary assembly using bwa mem 0.7.17-r1188[98], filtered out PCR duplicates using picard 2.27.1-0 (available at http://broadinstitute.github.io/picard), and scaffolded the assemblies using YaHS 1.1a[99]. We

ran BlobToolKit 2.6.5[100] on each scaffolded assembly and used the interactive web viewer to manually screen for scaffolds derived from non-target organisms. We removed no scaffolds from nxHelBake1, nxHelBake2, and nxHelBake3. We removed 95 scaffolds (representing 124 contigs) that had matches to rodents, *Giardia* spp., and bacteria from ngHelPoly1. The extent of host (*Apodemus sylvaticus*) and diplomonad (a *Giardia* spp. and a *Spironucleus* spp.) contamination in ngHelPoly2 meant that confidently identifying all contaminant scaffolds was not straightforward using BlobToolKit. To circumvent this, we used minimap2 2.24-r1122 to align the ngHelPoly2 read set to a concatenated FASTA file comprising our final, curated ngHelPoly1 assembly (see below), the *Apodemus sylvaticus* reference genome (GCA_947179515.1), and the *Giardia muris* reference genome (GCA_006247105.1) and discarded reads mapping to either *A. sylvaticus* or *G. muris*. We reassembled the remaining reads using hifiasm, scaffolded using YaHS, and ran BlobToolKit as previously described. We removed 39 remaining contaminant scaffolds (representing 38 contigs) from the ngHelPoly2 assembly. Following contamination removal, we extracted the mitochondrial genomes from each assembly using MitoHiFi 2.2[101]. Finally, we removed residual haplotypic duplication from each assembly using purge_dups 1.2.5[102] and used seqkit v2.1.0[103] and BUSCO 5.2.2[104] with the nematoda_odb10 lineage to calculate assembly metrics and assess biological completeness. We assessed base-level accuracy and *k*-mer completeness using Merqury 1.3[105].

## Hi-C scaffolding and manual curation

To generate chromosome-level reference genomes for *H. bakeri* and *H. polygyrus*, we selected the best assembly for each species based on numerical assembly metrics and biological completeness scores, which were nxHelBake1 and ngHelPoly1, respectively and scaffolded each assembly using Hi-C data as previously described. We performed manual curation of both scaffolded assemblies as in[106] and used PretextView 0.2.5 (available at https://github.com/wtsi-hpag/PretextView) to manually reorder, orient, break, and join scaffolds based on Hi-C signal. For nxHelBake1, we made 211 breaks, 358 joins, and removed 66 haplotigs. For ngHelPoly1, we made 218 breaks, 470 joins, and removed 24 haplotigs. Because the centre of the X chromosome is highly repetitive in both genomes, the contig order and orientation within these regions (X:37–55 Mb in nxHelBake1.1 and X:42–60 Mb in ngHelPoly1.1) is uncertain. We also found a heterozygous inversion on chromosome V in *H. bakeri* (V:53.0–60.6 Mb) with both orientations showing similar levels of Hi-C support. After curation, the six chromosome-sized scaffolds were reoriented to match each other and renamed using *C. elegans* nomenclature (I-V, X). Final Hi-C contact maps were generated using Juicer 2.0[107] and Juicebox 1.11.08 (available at https://github.com/aidenlab/Juicebox) and are shown in Fig. 1c and Supplementary Fig. 3A. To assess the GC-coverage bias of PiMmS, we aligned the Pacbio HiFi for nxHelBake1 and ngHelPoly1 to the respective reference genomes using minimap2. We calculated coverage in 100 bp windows using BEDtools v2.30.0[108] and mosdepth 0.3.3[109]. We calculated GC content in 100 bp windows using fasta_windows 0.2.4 (available at https://github.com/tolkit/fasta_windows). We filtered out any windows that were smaller than 100 bp or that were composed of ≥50% Ns.

## PacBio HiFi cDNA processing

To process the raw PacBio HiFi cDNA data, we followed the IsoSeq v3 processing pipeline (available at https://github.com/PacificBiosciences/IsoSeq/blob/master/isoseq-clustering.md) with an additional step to remove PCR duplicates. Briefly, we used lima v2.6.0 to demultiplex and remove Nextera adapter sequences and pbmarkdup v1.0.2 to remove PCR duplicates from the PacBio HiFi reads. We use IsoSeq refine 3.8.2 (https://isoseq.how/) to identify reads containing poly-A tails and trim them. We then clustered the refined read set into full-length transcripts using IsoSeq cluster 3.8.2 (https://

isoseq.how/) and assessed the biological completeness of each transcriptome using BUSCO.

## Repeat identification and curation

We generated initial de novo repeat libraries using earlGrey v1.2, with the nematoda Dfam library (Dfam 3.5), and 10 iterations of the "BLAST, Extract, Extend" process. We generated curated transposable element (TE) libraries for both *H. bakeri* and *H. polygyrus* following protocols described by[110]. We first excluded multicopy gene families (such as collagen, histones or sperm motility) from earlGrey repeat predictions, using pfamscan (http://ftp.ebi.ac.uk/pub/databases/Pfam/Tools/). Up to 50 TE copies of each remaining model were retrieved using the make_fasta_from_blast.sh script. We then aligned these sequences with MAFFT v7.487[111] and used trimAl v1.4. rev15[112] to remove poorly aligned positions and rare insertions (−gt 0.6). Alignments of each TE were manually inspected using AliView v1.28[113], to identify the start and end of each repeat. EMBOSS cons v6.6.0.0[114] was used to build the consensus sequence. In alignments where the edges were highly divergent, CIAlign 1.1.0 (available at https://github.com/KatyBrown/CIAlign) was implemented (--crop_divergent_min_prop_ident .5 --crop_divergent_buffer_size 15). TE-Aid (available at https://github.com/clemgoub/TE-Aid) was then used to detect structural features (such as terminal inverted repeats for DNA transposons and long terminal repeats for LTRs), to identify protein domains, and to split models where two or more transposons had been fused. Models that were split were then trimmed based on the coverage of the elements within individual models. Finally, we clustered redundant models using cd-hit-est v.4.8.1[115] with parameters suggested by Goubert et al. [110] and classified these TEs with the RepeatClassifier function from RepeatModeler2 v2.0.3[116]. The curated repeat libraries are available in the Zenodo repository.

## Protein-coding gene prediction

Prior to gene prediction, we soft-masked the nxHelBake1.1 and ngHelPoly1.1 reference genomes using RepeatMasker 4.1.2-p1 and the repeat libraries for each species. We first sought to predict genes using the BRAKER1 2.1.6 pipeline[117] by providing short-read RNA-seq data as evidence. For *H. bakeri*, we aligned six libraries of publicly available RNA-seq data[118]; (BioProject PRJNA486010) to the soft-masked nxHelBake1.1 reference genome using STAR 2.7.10a[119] and merged the resulting BAM files using SAMtools 1.3[120]. For *H. polygyrus*, we aligned three libraries of RNA-seq data generated from mixed sex adult nematodes (see above) to the soft-masked ngHelPoly1.1 reference genome using STAR and merged the resulting BAM files using SAMtools. The merged BAM files were provided to BRAKER1 to predict protein-coding genes in each reference genome.

We also independently predicted protein-coding genes using the BRAKER2 2.1.6 pipeline[121], which uses protein alignments from related species as evidence. To create a protein database, we downloaded genomes for 16 strongylid species and two outgroups (see Supplementary Table 6) from WormBase ParaSite (WBPS17)[122]. We ran BUSCO[104] (using the nematoda_odb10 lineage) on the 18 genomes downloaded from WormBase ParaSite and the nxHelBake1 and ngHelPoly1 reference genomes and used busco2fasta (available at https://github.com/lstevens17/busco2fasta) to identify and extract the protein sequences of 2887 BUSCOs that were single-copy in at least 4 of the 20 species. The resulting protein database, comprising 43,440 protein sequences, was provided to BRAKER2 to predict genes in the soft-masked nxHelBake1.1 and ngHelPoly1.1 genomes. We used TSEBRA v1.0.3[123] to combine the BRAKER1 and BRAKER2 predictions for both reference genomes, using a parameter set that favoured models derived from RNA-seq evidence over those derived from protein homology evidence (weight of 10000 for RNA-seq models and 1 for protein homology models) and which retained all ab initio models (i.e., those that did not have support from either RNA-seq or proteins). We

then used two rounds of PASA 2.5.2[124] to integrate the transcripts assembled with PacBio HiFi cDNA data.

To assess the quality of each gene set, we calculated numerical metrics using AGAT 0.8.1[125] (agat_sp_statistics.pl) and ran BUSCO (using the nematoda_odb10 lineages) on the predicted protein sequences to assess biological completeness. We also compared each gene set to the high-quality *H. contortus* gene set[20] by clustering the longest isoform of each predicted gene using OrthoFinder 2.5.4[126] and calculating the number of (a) single-copy orthologues (b) genes in orthogroups and (c) protein sequences whose length was within 10% of the *H. contortus* protein sequence.

### *H. mixtum* short-read genome assembly
We first removed adaptors and low-quality sequences from the PE short-read libraries for both individuals using fastp 0.23.2[127]. We estimated genome size and heterozygosity for each genome separately using JellyFish and GenomeScope2, as previously described. We then merged overlapping PE reads using VSEARCH v2.22.1[128], which lead to 83.8% and 79.3% of reads being successfully merged for Hm2 and Hm16, respectively. We assembled each set of merged reads independently using SPAdes 3.12.0[129] and assessed the results assembled using BUSCO and numerical metrics, as previously described. Finally, we identified and removed contamination from both assemblies using Blobtoolkit, as previously described. We confirmed species identity by extracting the ribosomal 18S sequences using hmmer and BEDtools (described below) and manually searched these against the 'nt' database using the NCBI BLAST server. Although both assemblies had low BUSCO completeness scores and were highly fragmented, we selected the Hm16 assembly for downstream analyses as it had marginally higher contiguity and biological completeness (Supplementary Table 7).

### Strongylid phylogeny
We downloaded genomes for 16 strongylid species and two outgroups (see Supplementary Table 6) from WormBase ParaSite (WBPS17)[122]. We ran BUSCO[104] (using the nematoda_odb10 lineage) on the 18 genomes downloaded from WormBase ParaSite and the nxHelBake1.1 and ngHelPoly1.1 reference genomes and used busco2fasta to identify and extract the protein sequences of 511 BUSCOs that were single-copy in at least 18 of the 20 species. We aligned the protein sequences using FSA 1.15.9[130] and inferred gene trees for each single-copy group using IQ-TREE v2.2.0.3[131] (allowing the best-fitting substitution model to be automatically selected[132]) and provided the resulting gene trees to Astral v5.7.8[133] to infer the species tree. To estimate branch lengths in amino acid substitutions per site, we concatenated the alignments of each single-copy orthogroup into a supermatrix using catfasta2phyml v1.1.0 (available at https://github.com/nylander/catfasta2phyml) and used IQ-TREE to estimate branch lengths under the general time reversible substitution model with gamma-distributed rate variation among sites. We visualised the resulting species tree using the iTOL web server[134]

### Nuclear ITS2 and mitochondrial trees
We downloaded the ribosomal RNA cistron ITS2 and mitochondrial cytochrome COI sequences used in[37] from NCBI GenBank, which comprised sequences from laboratory *H. bakeri* isolates, wild *H. polygyrus* isolates, and outgroups. To extract the ITS2 sequences from our single nematode assemblies, we aligned the NCBI ITS2 sequences using MAFFT v7.490 and used nhmmer v3.3.2[135] to construct a HMM from the alignment and search it against the genome assembly for each individual. We extracted the top hit for each assembly using BEDtools and combined these with the NCBI sequences before realigning with MAFFT. To obtain a COI alignment, we extracted the COI nucleotide sequences predicted by MitoFinder in the mitochondrial genome of each individual and aligned them with the NCBI COI

sequences using MAFFT. We then inferred ITS2 and COI trees using IQ-TREE, allowing the best-fitting substitution model to be automatically selected and 1000 ultrafast bootstraps[136]. We visualised the resulting trees using the iTOL webserver. To calculate average identity within and between species, we used a custom Python script (available at https://github.com/lstevens17/heligmosomoides_MS) to calculate pairwise divergence between each aligned ITS2 and COI sequence.

### Genome-wide divergence between *H. bakeri* and *H. polygyrus*
To estimate the average nucleotide identity between the *H. bakeri* and *H. polygyrus* genomes, we aligned the ngHelPoly1.1 genome to the nxHelBake1.1 genome using minimap2 (using the asm20 parameter). We calculated average identity by multiplying the length of each primary alignment by the gap-compressed per-base sequence divergence calculated by minimap2 and dividing by the total number of aligned bases.

To compare the divergence between *H. bakeri* and *H. polygyrus* to that between other closely related sister species pairs, we downloaded genomes and annotation files for *Brugia malayi*[137], *Brugia pahangi*[66], *Caenorhabditis briggsae*[138], *Caenorhabditis nigoni*[139], *Caenorhabditis remanei*[140], *Caenorhabditis latens*[141], *Onchocerca volvulus*[25], and *Onchocerca ochengi*[66] from WormBase Parasite (WBPS17). We used AGAT (agat_sp_keep_longest_isoform.pl) to remove non-longest isoforms from the protein files of all species and ran BUSCO (with the nematoda_odb10 lineage) on the isoform-filtered proteomes. For each species pair, we used busco2fasta to identify BUSCO genes that were single-copy in both proteomes. We aligned the protein sequences using FSA and translated protein alignments into codon alignments pal2nal v14[142]. We estimated $d_S$ for each pair of sequences using the Nei-Gojobori method implemented in codeml of PAML 4.9j[143] with runmode = −2. We filtered out orthologues that had $d_S \geq 0.8$ and calculated mean $d_S$ for each species pair.

### Divergence time estimation
We estimated the number of generations since *H. bakeri* and *H. polygyrus* last shared a common ancestor according to $T = (d_S - \pi_{anc})/(2\mu)$[144], where $T$ is equal to the number of generations, $d_S$ represents to the synonymous site divergence, $\pi_{anc}$ represents the nucleotide diversity at synonymous sites in the last common ancestor, and $\mu$ represents the per-site per-generation mutation rate.

Although extant *H. polygyrus* populations are likely a reasonable proxy for the last common ancestor of *H. bakeri* and *H. polygyrus* in terms of genetic diversity, we only sampled two *H. polygyrus* individuals from one local population meaning the nucleotide diversity in their genomes would like be an underestimate of the ancestral diversity. To generate a plausible estimate of nucleotide diversity at synonymous sites in the last common ancestor of *H. bakeri* and *H. polygyrus*, we instead took advantage of the species-wide resequencing dataset available for *H. contortus*[14], a globally distributed and genetically diverse parasite of sheep. We called variants in 34 *H. contortus* individuals (derived from 11 populations) with least 5x read coverage using BCFtools 1.15[145] and used VCFtools 0.1.16[146] to filter sites where over 80% of individuals had missing data. We identified four-fold degenerate sites in the *H. contortus* reference genome using degenotate 1.2.4 (available at https://github.com/harvardinformatics/degenotate) and used pixy 1.2.7.beta1[147] to calculate synonymous site diversity in each population. We calculated the mean synonymous site diversity in the 11 *H. contortus* populations to be 3.2%.

Using the *C. elegans* mutation rate of $2.7 \times 10^{-9}$ per site per generation[148], we estimate that *H. bakeri* and *H. polygyrus* last shared a common ancestor ~6.36 million generations ago. Although *H. bakeri* and *H. polygyrus* have a generation time of ~16 days[46], the number of generations completed each year is likely to be substantially lower than the theoretical maximum of 22.8 generations per year. Field-based estimates for related strongylids range from one to six

generations per year[149–153]. Therefore, 6.36 million generations corresponds to between 1.06 million years ago (assuming an average of 6 generations per year) and 6.36 million years ago (assuming an average of 1 generation per year). We stress that these estimates are highly uncertain as they rely heavily on the following assumptions: (a) that present-day *H. contortus* populations contain similar levels of diversity to those in the last common ancestor of *H. bakeri* and *H. polygyrus*, (b) that 1-6 generations per year is a plausible range for the average number of generations per year in the *Heligmosomoides* species, and (c) that the *C. elegans* mutation rate is similar to that of *Heligmosomoides* species.

## SNP calling and analysis

We used minimap2 to align PacBio HiFi reads for each *H. bakeri* and *H. polygyrus* individual to the nxHelBake1.1 and ngHelPoly1.1 reference genomes, respectively. We provided the BAM files to DeepVariant 1.4.0[154] to call variants and filtered the VCF to remove any variant labelled with 'RefCall'. We used BCFtools 1.15[145] to filter the resulting VCF files to contain only heterozygous biallelic single nucleotide variants (SNP) that had QUAL values ≥15. To avoid our analyses of heterozygous SNP density and distribution being biased by the highly repetitive nature of the genomes, we remove any SNPs that overlapped with repeat annotations and calculated SNP density in 10 kb windows as the number of non-repetitive SNPs divided by the number of non-repetitive bases. To calculate genome-wide average heterozygous SNP density in nxHelBake1 (which was male and therefore had a hemizygous X chromosome), we divided the span of the non-repetitive portion of the autosomal genome by the number of SNPs that were called in those regions. As ngHelPoly1 was female and therefore had two copies of the X chromosomes, we calculated SNP density using all non-repetitive portions of the genome.

## Hyper-divergent haplotype calling

Hyper-divergent haplotypes can be identified in whole-genome alignments as regions that contain low identity (and often short) alignments that are flanked by high identity alignments. In addition, they also typically contain higher numbers of SNPs than the genome-wide average. We developed a pipeline to identify hyper-divergent haplotypes in each of the non-reference haplotypes (nxHelBake1 alternate, nxHelBake2 primary, nxHelBake2 alternate, nxHelBake3 primary, nxHelBake3 primary) based on these characteristics. First, we aligned each haplotype to the nxHelBake1.1 reference genome using NUCmer 3.1[155] and removed all alignments that were not one-to-one alignments using delta-filter. We used BEDtools to identify gaps between alignments of a given length and identity. However, homozygous regions are not represented in the alternate assemblies generated by hifiasm, meaning alignment gaps could either represent regions of high divergence or homozygous regions. To differentiate between these two, we called SNPs in each individual by aligning each haplotype assembly to the nxHelBake1.1 reference genome using minimap2 and called variants using paftools 2.24-r1122 (allowing for a minimum alignment length of 1000 bp). We calculated SNP density in each alignment gap using BEDtools as described previously.

To identify the optimal parameter set to call hyper-divergent haplotypes, we ran the same pipeline on PacBio genome assemblies of 15 strains of *C. elegans* used by Lee et al.[17] using the *C. elegans* N2 (version WB286) genome as the reference genome across a range of parameter values. We then identified the overlap between the hyper-divergent region coordinates defined by Lee et al.[17] and those generated by our approach using BEDtools coverage. We calculated sensitivity as the number of overlapping bases divided by the total number of bases defined by Lee et al.[17] and specificity as the number of overlapping bases divided by the total number of bases classified by our approach (Supplementary Fig. 11). We chose the optimal parameter set as any gap that was ≥10 kb in length between flanking alignments that

had ≥95% identity and that were ≥2 kb in length and that had a SNP density of ≥0.002. This parameter set classified 3.72 Mb of the 4.56 Mb defined by Lee et al.[17] as hyper-divergent across all 15 strains (sensitivity of 81.3%) and classified 4.69 Mb in total (specificity of 79.1%).

Finally, we ran the hyper-divergent haplotype calling pipeline using this parameter set on the five non-reference *H. bakeri* haplotype assemblies to classify divergent regions. To avoid characterising homozygous regions as hyper-divergent on the basis of short, misaligned sequences, we applied an additional filter: at least 30% of any hyper-divergent haplotype should be covered by 1 kb bins that contain SNPs and by NUCmer alignments of any identity. Additionally, we removed any hyper-divergent haplotypes that were called on the X chromosome in the three alternate haplotypes (we removed 2, 1, and 1 X chromosome calls from the nxHelBake1, nxHelBake2, and nxHelBake3 alternate assemblies, respectively) and any hyper-divergent haplotype that was called on the non-chromosomal scaffolds. We used BEDtools intersect to identify protein-coding genes where ≥50% of their length was covered by a hyper-divergent haplotype.

## GO term analysis

We functionally annotated the longest isoform of each predicted protein-coding gene using InterProScan 5.54–87.0[156]. The resulting GO terms for each protein were supplied to the TopGO R package 2.50.0[157] along with a list of all hyper-divergent genes. We ran GO term enrichment using all three GO categories ('molecular function', 'biological process', and 'cellular component') using the 'weight01' algorithm, which takes the GO hierarchy into account when calculating enrichment. We only considered GO terms that were significant ($p$-value < 0.05) using 'weight01' and Fisher's exact test.

## Identifying haplotype sharing between *H. bakeri* and *H. polgyrus*

To identify haplotype sharing between *H. bakeri* and *H. polygyrus*, we first used miniprot 0.7-r20[158] and the longest isoform of each protein-coding gene in the nxHelBake1.1 reference to predict genes all five non-reference *H. bakeri* haplotype assemblies, all four *H. polygyrus* haplotype assemblies, and the *H. mixtum* assembly. We filtered the miniprot predictions to include only the top-ranked prediction per protein and removed any predictions that had more than one stop codon or that had a frameshift. We extracted the protein sequences from the filtered GFF3 files using AGAT. For every hyper-divergent gene, we extracted the predicted protein sequence from all *H. bakeri* haplotypes where that gene was classified as hyper-divergent, along with the predicted protein sequences in all *H. polygyrus* haplotypes and in *H. mixtum*. We filtered any groups that lacked either a *H. polygyrus* or *H. mixtum* representative and that had only one *H. bakeri* representative. We aligned the remaining groups of protein sequences using FSA and inferred gene trees using IQ-TREE using the LG substitution model with gamma-distributed rate variation among sites (+Γ) and 1000 ultrafast bootstraps. We then used a custom Python script (available at https://github.com/lstevens17/heligmosomoides_MS) and the ete3 Python module[159] to root gene tree using the *H. mixtum* protein sequence as the outgroup and then traverse the unrooted tree to find clades that contained both *H. bakeri* and *H. polygyrus* sequences (i.e., gene trees where the *H. bakeri* and *H. polygyrus* sequences did not form two separate clades) that had bootstrap support of 50 or greater. To prevent mispredicted or misaligned sequences from influencing our results, we calculated the median distance between the *H. mixtum* sequence and the *H. bakeri* and *H. polygyrus* sequences in each gene tree and removed any sequences whose distance to *H. mixtum* was ≥2 times the median distance.

## Comparing divergence within shared haplotypes to the genome-wide average

To compare the levels of divergence between hyper-divergent haplotypes that are shared between nxHelBake1 and ngHelPoly1 to the

 

genome-wide average, we identified gene trees where the clade comprising a mixture of *H. bakeri* and *H. polygyrus* sequences contained both nxHelBake1 primary and ngHelPoly1 primary. We then inferred single-copy orthologues between the nxHelBake1.1 and ngHelPoly1.1 reference genomes using OrthoFinder and aligned the proteins sequenced using FSA. We calculated $d_S$ as previously described. We then compared mean $d_S$ between single-copy orthologues found within shared haplotypes ($N = 189$) and all orthologues found in non-divergent regions of the *H. bakeri* genome ($N = 9753$).

### Reporting summary

Further information on research design is available in the Nature Portfolio Reporting Summary linked to this article.

## Data availability

The *H. bakeri* nxHelBake1.1 and *H. polygyrus* ngHelPoly1.1 reference genomes have been deposited in ENA under the BioProject accessions PRJEB57615 and PRJEB57641, respectively. Accession numbers for the other assemblies generated as part of this study (alternate assemblies and assemblies for the other three individuals) are available in Supplementary Table 8. PacBio and Hi-C data have been deposited in ENA under the BioProject accessions PRJEB46574 and PRJEB36817. The *H. mixtum* raw data and genome assembly have been deposited in ENA under the BioProject PRJEB61185. The *H. bakeri* RNA-seq reads used in gene prediction are available in ENA under the BioProject PRJNA486010. The *H. polygyrus* short-read RNA-seq reads have been deposited in ENA under the BioProject PRJEB61184. Accession numbers for the genome assemblies and annotations used for gene predictions and phylogenomic analysis are available in Supplementary Table 6. Large data files associated with this manuscript, including VCFs, gene annotation files, curated repeat libraries, and Newick files have been deposited in Zenodo at https://doi.org/10.5281/zenodo.8403377 [https://zenodo.org/doi/10.5281/zenodo.8403377]. Data files associated with the manuscript can be found in the GitHub (https://github.com/lstevens17/heligmosomoides_MS), which has been accessioned in Zenodo at https://doi.org/10.5281/zenodo.10092962. Source data associated with the figures and tables can be found in the Source Data file, the GitHub repository, and in ENA. Source data are provided with this paper.

## Code availability

Code associated with the analyses and figures is available via GitHub (https://github.com/lstevens17/heligmosomoides_MS), which has been accessioned in Zenodo at https://doi.org/10.5281/zenodo.10092962.

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

## Acknowledgements

This research was funded in part by the Wellcome Trust Grants 206194 and 218328 awards to the Wellcome Sanger Institute and Leverhulme grant RPG-2019-404 awarded to A.H.B. For the purpose of Open Access, the authors have applied a CC BY public copyright licence to any Author Accepted Manuscript version arising from this submission. We are grateful to the Sanger Scientific Operations Long Read Team and the

Genome Reference Informatics Team (GRIT) for their assistance with sequencing and curation and to Tree of Life colleagues for reading and commenting on an earlier draft of this manuscript. We thank Daniel Fusca and Asher Cutter for their advice on calculating synonymous site diversity.

## Author contributions

L.S., A.B.P., C.A.-G., A.H.B., and M.B. conceptualised the project. R.B., J.L.H., A.K., and E.R. collected the parasite material. L.S., E.K., P.G., J.L.H., M.K., A.K., and E.R. generated the sequencing data. L.S., I.M.-U., and M.W. analysed the data. D.A. and S.P. manually curated the reference genomes. A.B.P., C.A.-G., A.H.B., and M.B. supervised the project. L.S. drafted the manuscript. All authors commented on, revised, and approved the manuscript.

## Competing interests

The authors declare no competing interests.
