## [Peer Review File · Nature Communications]

Ancient diversity in host-parasite interaction genes in a model parasitic nematodeREVIEWER COMMENTS

Reviewer #1 (Remarks to the Author):

The manuscript "Ancient diversity in host-parasite interaction genes in a model parasitic nematode" by Stevens et al. presents a comparative genomic study of two parasitic nematodes, *Heligmosomoides bakeri* and *H. polygyrus*. The authors generated high-quality genomes of several individuals from these species, demonstrating a high level of genome divergence and confirming that *H. bakeri* and *H. polygyrus* are two distinct species. They identified numerous hyper-divergent haplotypes and associated some of them with host-parasite interactions. Then the authors delved deeper and discovered several remarkable examples of trans-specific polymorphisms that have been under long-term balancing selection.

The manuscript is well-written, easy to follow, and well-structured. The authors employed cutting-edge methods, which they described in detail. Their approach has the potential to facilitate the study of parasitic nematodes, as well as other species that are small in size and contain only picograms of DNA. I enjoyed reading this work and appreciated the detailed discussion of the potential functions of some haplotypes and all the additional phylogenetic analyses they conducted. I believe that this study will be of great interest to the broad audience of Nature Communications.

I have a few minor suggestions that could further improve the manuscript.

Comments:

1. Lines 224 and 228: Given that ~64% of the genomes are masked (line 175), could you please elaborate on what percentage of non-repetitive regions in each species aligned with another species? I'm slightly confused how it can be 56% of the total length, as that would include some repeats as well.
2. The authors discussed local gaps in the read coverage within hyper-divergent haplotypes (line 315). Were there many cases with, on the contrary, local high-coverage peaks within divergent haplotypes? Did you use any additional filters for those haplotypes?
3. Please comment on the likely reason for SNP density peaks and inferred hyper-divergent regions on hemizygous X chromosomes (Figures 3B and S10)
4. Did you use any additional filters for minimap2 alignments before variant/haplotype calling? If yes, please add this information to the Methods section.
5. Line 738: a typo in "picard".
6. Please provide the version of WormBase ParaSite used in this study

Reviewer #2 (Remarks to the Author):

Stevens et al. presented three main major part in this study: i) by using the PiMmS protocol that the PCR amplified minute amount of genomic DNA, the authors were able to sequence and produce assembly from HiFi data from single parasitic nematodes which the authors considered challenging to retrieve from animal hosts, ii) further scaffolded using HiC data resulting in chromosome level assemblies, and based on these genomes iii) delimit *Heligmosomoides bakeri* and *Heligmosomoides polygyrus* as distinct biology and reveal hyper-divergent haplotypes that have persisted since their last common ancestor. The authors went on and show that these haplotypes were enriched with gene families associated with host immune response.

I have a few major questions about the scientific points in the paper, along with minor comments

Major comments

The first major advance is the PiMmS protocol, which provides the needed technical advances enabling the sequencing and assembly of single nematodes. However, the details and potential bias (or no bias) of this protocol was totally lacking. This protocol seems to rely on long-range PCR amplification, so it is curious to find out the PCR duplicate % and the amplification bias such as going across GC content. The former was nicely shown in Figure S1 but there was no indication in the latter. The authors have stated in L477 "When used to amplify whole genomes, PCR is known to introduce coverage biases across the genome (Pinard et al. 2006), which can lead to coverage dropouts and therefore assembly gaps". The authors should plot coverage across the genome to visualise the evenness. To live up to its claims, which the subsequent part of the study heavily rely on, the authors are recommended to spend greater efforts in characterizing the capability of PiMmS. Currently, it is left without any support. Example include L489 "Importantly, the PiMmS protocol has been used to sequence nematodes as small as *C. elegans* (an adult *C. elegans* hermaphrodite is 1 mm long⁴⁹¹ and contains ~200 pg of DNA), meaning this protocol will likely be applicable to even the smallest parasitic nematodes". Is there literature to support this?

The second relates to the confidence of mapping. This relies heavily on two aspects of the data: i) sequencing coverage and evenness and ii) genome mappability. It is unclear from Figure 3 how much of the pattern were genuine or caused by the two factors. For instance, the center of the X chromosome are highly repetitive, yet the heterozygosity are very low in both species. A coverage of non-overlapping window across the chromosomes, as well as the repeat over these plot would give the authors some indication on the confidence. A similar point of mapping confidence would be how many of these reads were correctly mapped to their designated positions. For example, Figure 3C nicely shows reads were supporting divergent haplotype around 87,740 kb because the reads have both the "hyper" and "similar" portion that span across the "hyper" part. In contrast, all reads around 87,770 kb do not and they were simply mapping to the most plausible position in the assembly without additional support.

The third relates to balancing selection maintaining "Ancient genetic diversity in genes associated with host-parasite interactions". In order to claim i) balancing selection is taking place and ii) some of the alleles were already present in the last common ancestor, the more stringent way is to identify an allele (gene) that has five versions: the *Heligmosomum mixtum* allele, the *H. bakeri* (primary and alternative) version, and the *H. polygyrus* (primary and alternative) version. For the hyper haplotype to be present since the split, the tree must look like. (*Heligmosomum mixtum* allele(*H.bakeri* allele 1, *H. polygyrus* allele 1)),(*H.bakeri* allele 2, *H. polygyrus* allele 2). The readers will feel convinced if a sufficient number of alleles are present in the assemblies.

Other comments

L36 The results were not unexpected. There are plenty of literatures that have documented the presence of balancing selection blocks including the ones introduced (for instance Doyle et al 2020) and published by the authors (for instance Stevens et al 2022).

L76-86 This is not true. There are now many reference genomes that are produced outside the Sanger Institute that the authors failed to recognize. Looking at WormBase ParaSite (WBPS) and sort by N50, there are some plant parasitic nematodes that are arguably also very important that have the assembly in chromosomes. Even beyond the database, assembly of *Ascaris suum* (Wang et al 2020 Current Biology) and *H. glycines* (Masonbrink et al 2021 Mol. Ecol. Res) are now in chromosomes also important resources. The authors should acknowledge these efforts and that parasite genomes are being revamped with reference genomes.

L79 The ultra-low Pacbio kit only require 5ng of genomic DNA, which have been routinely sequenced in single insects. The authors should acknowledge this advancement. Note: The ultra-low Pacbio was referred as "similar" protocol in L483. Since the PiMmS could sequence with fewer gDNA, the authors are recommended to discuss in details.

L125-126 The read N50 from 9.6-11.1kb was quite low to standard HiFi data, which may be due to the original gDNA quality or the PiMmS protocol. Please explain in more details.

Figure 1B It is not known how the cumulative contig plot of 1B shows how the variations in assembly metric were largely driven by read coverage.

Figure 1B Type PRJEB15393 in legend.

L176 Please explain source of short-read RNA-seq.

L186 Please explain "due to technical reasons rather than biological differences". Do the 1,505 more genes in *H. polygyrus* were left unannotated in *H. bakeri* die to "technical reasons" or they were false positives?

Table 1 some details should be spent on describing the almost ~50Mb difference in *H. bakeri* WBPS17 and nxHelBake1.1. Is it to do with phased assembly or PCR bias in the PiMmS?

L232-234. A figure is needed to show the inversion as well as repeat content across the X chromosome in both species. This will help to understand the apparent lack of points in the middle of X chromosome in Figure 2D and S6A.

L243 The authors have referred that the two species have diverged million of years ago in several places of the manuscript so it would be good to put a rough estimate here, as opposed to refer the *C. briggsae* and *C. nigoni* divergence and left the readers wondering

Figure S7 C) label should be *H. bakeri* nxHelBake3

Reviewer #3 (Remarks to the Author):

Heligmosomoides bakeri and *Heligmosomoides polygyrus* are parasitic nematodes of rodents that are widely used laboratory models because of their phylogenetic proximity to human hook worms. This paper reports chromosome-level assemblies of these two *Heligmosmoides* species. What makes this study valuable is that they used single-worm long-read sequencing to generate assemblies and

compare the genomes of multiple individuals of these species, which has not been previously reported in other parasitic nematodes. It should also be noted that they successfully constructed chromosome-level assemblies from single individuals using pooled DNA HiC information. Long-read sequence data from multiple individuals were used for haplotyping and detection of hyper-variable regions. The genome comparison revealed the phylogenetic relationship of the two species, which were previously unclear as to whether or not they were the same species. The haplotype information also showed that there is 'haplotype sharing' between the two species, thought to be the result of blanching selection. These regions have been maintained through long periods of inbreeding in the laboratory and appear to be important for parasitism. The study also suggests that the worms used in laboratories are likely to be genetically different between laboratories, despite having the same origin, and warns that phenotypic analyses can vary between laboratories.

The genome sequencing and informatics analysis were beautifully done and the paper is logical and well written. Unfortunately, the paper does not include any experimentally based data, such as phenotypic observations followed by gene disruption, but I think the novelty described at the beginning of the paper is sufficient to compensate for this.

Minor points;

Line 168; 124,254 and 45,333 what are these numbers for? Probably they are full length transcripts of *Heligmosomoides bakeri* and *Heligmosomoides polygyrus*, respectively. Also any idea why the numbers are so different from each other.

Line 472; "contigs" instead of "assemblies"?

Line 500-505; redundant with the introduction. Please reduce text amount.

Line 542; that have previously reported to be associated with?

Response to reviewers

Reviewer comment

Author response

Reviewer 1

The manuscript "Ancient diversity in host-parasite interaction genes in a model parasitic nematode" by Stevens et al. presents a comparative genomic study of two parasitic nematodes, Heligmosomoides bakeri and H. polygyrus. The authors generated high-quality genomes of several individuals from these species, demonstrating a high level of genome divergence and confirming that H. bakeri and H. polygyrus are two distinct species. They identified numerous hyper-divergent haplotypes and associated some of them with host-parasite interactions. Then the authors delved deeper and discovered several remarkable examples of trans-specific polymorphisms that have been under long-term balancing selection.

The manuscript is well-written, easy to follow, and well-structured. The authors employed cutting-edge methods, which they described in detail. Their approach has the potential to facilitate the study of parasitic nematodes, as well as other species that are small in size and contain only picograms of DNA. I enjoyed reading this work and appreciated the detailed discussion of the potential functions of some haplotypes and all the additional phylogenetic analyses they conducted. I believe that this study will be of great interest to the broad audience of Nature Communications.

We thank the reviewer for their positive feedback on our manuscript.

I have a few minor suggestions that could further improve the manuscript.

Comments:

1. Lines 224 and 228: Given that ~64% of the genomes are masked (line 175), could you please elaborate on what percentage of non-repetitive regions in each species aligned with another species? I'm slightly confused how it can be 56% of the total length, as that would include some repeats as well.

The reviewer is correct that a substantial proportion of the sequence that is aligned between the nxHelBake1.1 and ngHelPoly1.1 genomes overlaps with our repetitive (i.e. transposable) element annotations. Using a MAPQ threshold of 60 (the highest reported by minimap2), 363 Mb (55%) of nxHelBake1.1 genome is covered by a *H. polygyrus* alignment, 212 Mb (58%) of which overlaps with repetitive elements. For example, there is a 114 kb alignment on the centre of *H. bakeri* chromosome

IV (IV:51,723,204-51,837,803) that has an average identity of 98.2%. Within this 114 kb region, there are 218 annotated repetitive elements, which span 77.5 kb (68% of the aligned region). We are able to align across repetitive elements because the contigs are far longer than individual repetitive elements and so the alignments can be anchored by the unique flanking sequences. If we opted to exclude all regions that contained a transposable element, 69.6% of the remaining genome (representing approximately 1/3 of the overall genome) is covered by an alignment. However, this is a substantially biased estimate of the alignable portion of the two genomes because both repeat density and divergence rates vary across the genome. We therefore believe it makes more sense to include regions containing repetitive element annotations when estimating divergence between the *H. bakeri* and *H. polygyrus* genomes.

2. The authors discussed local gaps in the read coverage within hyper-divergent haplotypes (line 315). Were there many cases with, on the contrary, local high-coverage peaks within divergent haplotypes? Did you use any additional filters for those haplotypes?

During the development of our hyper-divergent haplotype calling pipeline, we manually inspected a large number of haplotype calls using IGV. As the reviewer suggests, misaligned reads, particularly in repetitive regions, can lead to local high-coverage peaks which, if not filtered, could make non-divergent/homozygous regions appear as if they contain many variants. Although we did not use information from read alignments (such as coverage) in our pipeline, the same issues could, in theory, affect whole-genome alignments. To prevent these regions from being called as hyper-divergent by our pipeline, we applied two key filters:

1. We used delta-filter (part of the MUMmer package) to retain only one-to-one NUCmer alignments prior to hyper-divergent haplotype calling. This is noted in our methods section as *“First, we aligned each haplotype to the nxHelBake1.1 reference genome using NUCmer 3.1 (Kurtz et al. 2004) and removed all alignments that were not one-to-one alignments using delta-filter.”*
2. We added a filter to explicitly ignore regions where only a small proportion of the region was covered by variants or non-repetitive alignments. This is described in our methods section as *“To avoid characterising homozygous regions as hyper-divergent on the basis of short, misaligned sequences, we applied an additional filter: at least 30% of any hyper-divergent haplotype should be covered by 1 kb bins that contain SNPs and by NUCmer alignments of any identity”*.

Manual inspection of the calls after applying these additional filters suggested that they were effective at preventing non-divergent/homozygous regions containing misaligned sequences from being called hyper-divergent.

3. Please comment on the likely reason for SNP density peaks and inferred hyper-divergent regions on hemizygous X chromosomes (Figures 3B and S10)

As the reviewer implies, the heterozygous SNPs on the X chromosome for those individuals that were male (nxHelBake1, nxHelBake2, nxHelBake3, and ngHelPoly2) must be errors because the X was hemizygous in these individuals. In contrast, ngHelPoly1, shown in Figure 3A, was female and so had genuine heterozygosity on the X chromosomes. To understand why peaks existed on the X chromosomes of male individuals, we used IGV to manually inspect windows on the X that had high SNP density and found that most appear to be due to reads being misaligned to repetitive regions (which were presumably missed by our repeat calling pipeline). For example, the outlying 10 kb window in nxHelBake1 (Figure 3B) in the middle of the X (X:53600000-53610000) has a SNP density of 0.026, which is derived from misaligned reads (as indicated by the sudden increase in coverage; see IGV screenshot below).

IGV view of a region on the nxHelBake1.1 X chromosome (X:53600000-53610000) that contains a high number of heterozygous SNPs.

Fortunately, windows with high SNP densities are rare on the X chromosome. Using nxHelBake1 as an example, only 25 windows on the X (spanning 250 kb, 0.27% of total chromosome) have SNP densities of ≥ 0.01 (equivalent to 1 SNP every 100 bp). In contrast, the number of windows with SNP densities of ≥ 0.01 on the autosomes ranges from 241 (2.4 Mb) on chromosome III to 1,360 (13.6 Mb) on chromosome V. Overall, only 3,270 (0.89%) of the 367,123 heterozygous SNPs called in non-repetitive regions in nxHelBake1 are on the X chromosome.

Importantly, our hyper-divergent haplotype calls are not significantly impacted by erroneous X SNPs. As hyper-divergent haplotypes were called relative to the nxHelBake1.1 reference genome, the haplotypes called on the X chromosome in the primary haplotypes of nxHelBake2 and nxHelBake3

are real (and represent regions where the hemizygous X in these individuals is hyper-divergent from the X in nxHelBake1). In contrast, any hyper-divergent region called in the alternate haplotypes of any of the three *H. bakeri* individuals are erroneous (because the X is hemizygous). Reassuringly, our pipeline only found a very small number of these: two of the 685 haplotypes called in nxHelBake1 alternate were on the X, one (out of 469) in nxHelBake2 alternate, and one (out of 154) in nxHelBake3 alternate. These were removed prior to downstream analysis. This is noted in our methods section as *“Additionally, we removed any hyper-divergent haplotypes that were called on the X chromosome in the three alternate haplotypes (we removed 2, 1, and 1 X chromosome calls from the nxHelBake1, nxHelBake2, and nxHelBake3 alternate assemblies, respectively)”*.

To ensure the readers aren't puzzled as to why there are SNP density peaks on the X chromosomes of male individuals, we have added *“<ToLID> was male and therefore had a hemizygous X chromosome; the SNP density peaks on the X chromosome are therefore erroneous and are derived from mismapped PacBio HiFi reads”* to the legends of Figure 3 and Figure S9.

4. Did you use any additional filters for minimap2 alignments before variant/haplotype calling? If yes, please add this information to the Methods section.

The minimap2 alignments were not directly filtered prior to variant calling, but we did set the minimum alignment length in pafutils.js to 1000 bp when calling variants. We have added this to our methods. For hyper-divergent haplotype calling, we used NUCmer as the whole genome alignment software rather than minimap2 so that we could use delta-filter (part of the MUMmer package) to filter out repetitive alignments. This is noted in the methods as *“First, we aligned each haplotype to the nxHelBake1.1 reference genome using NUCmer 3.1¹⁵⁵ and removed all alignments that were not one-to-one alignments using delta-filter”*.

5. Line 738: a typo in "picard".

We have replaced 'piccard' with 'picard'.

6. Please provide the version of WormBase ParaSite used in this study

We have added WormBase ParaSite version (17) to the relevant methods sections and to Table S7, which describes accessions/versions of genomes used in our study.

Reviewer 2

Stevens et al. presented three main major part in this study: i) by using the PiMmS protocol that the PCR amplified minute amount of genomic DNA, the authors were able to sequence and produce assembly from HiFi data from single parasitic nematodes which the authors considered challenging to retrieve from animal hosts, ii) further scaffolded using HiC data resulting in chromosome level assemblies, and based on these genomes iii) delimit Heligmosomoides bakeri and Heligmosomoides polygyrus as distinct biology and reveal hyper-divergent haplotypes that have persisted since their last common ancestor. The authors went on and show that these haplotypes were enriched with gene families associated with host immune response.

I have a few major questions about the scientific points in the paper, along with minor comments

Major comments

The first major advance is the PiMmS protocol, which provides the needed technical advances enabling the sequencing and assembly of single nematodes. However, the details and potential bias (or no bias) of this protocol was totally lacking. This protocol seems to rely on long-range PCR amplification, so it is curious to find out the PCR duplicate % and the amplification bias such as going across GC content. The former was nicely shown in Figure S1 but there was no indication in the latter. The authors have stated in L477 "When used to amplify whole genomes, PCR is known to introduce coverage biases across the genome (Pinard et al. 2006), which can lead to coverage dropouts and therefore assembly gaps". The authors should plot coverage across the genome to visualise the evenness. To live up to its claims, which the subsequent part of the study heavily rely on, the authors are recommended to spend greater efforts in characterizing the capability of PiMmS. Currently, it is left without any support. Example include L489 "Importantly, the PiMmS protocol has been used to sequence nematodes as small as C. elegans (an adult C. elegans hermaphrodite is 1 mm long and contains ~200 pg of DNA), meaning this protocol will likely be applicable to even the smallest parasitic nematodes". Is there literature to support this?

While we do not provide a detailed analysis of PiMmS in our manuscript, we respectfully disagree that the protocol is left without any support. The genome assemblies themselves provide clear evidence that PiMmS is capable of producing high-quality genomes from single specimens. We present extensive QC in our manuscript to show that, where we obtained sufficient coverage, our PiMmS-derived genomes have high gene completeness, high contiguity, and high base-level accuracy. Importantly, their quality surpasses that of the majority of previously published parasitic nematode reference genomes.

However, we do agree with the reviewer that understanding the biases and limitations of the PiMmS protocol is important. Dr Chris Laumer, who developed the PiMmS protocol, is currently preparing a manuscript that analyses in detail how it performs on individual *C. elegans* hermaphrodites. In contrast to our *Heligmosomoides* data, the *C. elegans* PiMmS data can be compared to a telomere-to-telomere reference genome derived from non-amplified DNA. Given that this manuscript is yet to be published, we agree with the reviewer that it is important to understand the PCR duplicate rates and coverage biases in our *Heligmosomoides* data. We provide PCR duplicate rates for all five individuals in Table S1. To provide some insight into the GC bias of PiMmS, we have added a new supplementary figure (Figure S5; which we have inserted below) to show the relationship between coverage and GC in 100 bp windows across the nxHelBake1.1 and ngHelPoly1.1 reference genomes. This shows that data from PiMmS displays relatively little coverage bias across GC values. Comparing these plots to those shown in PacBio's Low and Ultra Low Input method application note (see <https://www.pacb.com/wp-content/uploads/Application-Note-Considerations-for-Using-the-Low-and-Ultra-Low-DNA-Input-Workflows-for-Whole-Genome-Sequencing.pdf>), the pattern is qualitatively very similar to that shown for the PacBio low input protocol (which does not use PCR amplification) and shows substantially less bias than the PacBio ultra-low input protocol (which does use PCR amplification). We reference this new supplementary figure in the results by saying "*Although our data show relatively little bias across GC values (Figure S5), we note that the completeness score of our H. bakeri reference genome is slightly lower than the previous H. bakeri reference genome⁴³ (94.3%) which may be due to coverage dropouts associated with long-range PCR amplification*". We have also updated our discussion and methods sections to reflect this new analysis.

As mentioned above, the PiMmS *C. elegans* results have not yet been published. We have therefore added "(Laumer CE, pers. comm.)" to that sentence.

Figure S5: GC-coverage bias in the *H. bakeri* nxHelBake1 and *H. polygyrus* ngHelPoly1 PIMmS data

PacBio HiFi reads coverage in 100 bp windows across the (A) *H. bakeri* nxHelBake1 and (B) *H. polygyrus* ngHelPoly1 reference genomes, binned by GC %. Histograms show the counts of windows in each bin. Only windows that were 100 bp in length and that had $\leq 50\%$ Ns are shown. No windows in either genome had a GC% of $> 90\%$.

The second relates to the confidence of mapping. This relies heavily on two aspects of the data: i) sequencing coverage and evenness and ii) genome mappability. It is unclear from Figure 3 how much of the pattern were genuine or caused by the two factors. For instance, the center of the X chromosome are highly repetitive, yet the heterozygosity are very low in both species. A coverage of non-overlapping window across the chromosomes, as well as the repeat over these plot would give the authors some indication on the confidence. A similar point of mapping confidence would be how many of these reads were correctly mapped to their designated positions. For example, Figure 3C nicely shows reads were supporting divergent haplotype around 87,740 kb because the reads have both the "hyper" and "similar" portion that span across the "hyper" part. In contrast, all reads around 87,770 kb do not and they were simply mapping to the most plausible position in the assembly without additional support.

We agree with the reviewer that reads often map to incorrect locations and that it is important to understand how mismapping might influence our results. However, our PacBio HiFi reads have an average QV above 30 (corresponding to an error rate of less than 1 in 1000) and a read N50 of ~10 kb. The error rate associated with mapping PacBio HiFi data is therefore orders of magnitude lower than associated with the more commonly-used Illumina data (which have lengths 100-250 bp), especially in repetitive regions.

We did, however, take steps to prevent mismapped reads (and misaligned regions in our whole genome alignments) from influencing our results:

- We filtered out any heterozygous SNPs that were called in repetitive elements (thus removing over half of all heterozygous SNPs called by deepvariant)
- We filtered NUCmer alignments to contain only one-to-one alignments prior to hyper-divergent haplotype calling
- We explicitly ignored hyper-divergent haplotype calls where only a small proportion of the region was covered by variants or non-repetitive alignments

With regards to heterozygosity on the centre of the X chromosomes, the only individual with genuine heterozygosity on the X is ngHelPoly1 (which was female; all other individuals were male and therefore have hemizygous X chromosomes). The reason that heterozygosity appears to be low in the centre of the X chromosome in ngHelPoly1 is because we filtered out SNPs called in repetitive regions (and the centre of X is highly repetitive) rather than because of our ability to map reads.

Given that the X chromosome is hemizygous in males, any heterozygous SNPs called on the X chromosome in the male individuals are erroneous. This provides us with an opportunity to estimate the false positive rate associated with our SNP calling. As noted in our response to reviewer #1, only

3,270 (0.89%) of the 367,123 heterozygous SNPs called in non-repetitive regions in nxHelBake1 are on the X chromosome.

In summary, the mismapping rate in our analysis is substantially lower than that associated with the far more commonly-used Illumina reads, we took several steps to avoid misaligned sequences from influencing our results, and mismapped reads have had a negligible impact on the results we present in our manuscript.

The third relates to balancing selection maintaining "Ancient genetic diversity in genes associated with host-parasite interactions". In order to claim i) balancing selection is taking place and ii) some of the alleles were already present in the last common ancestor, the more stringent way is to identify an allele (gene) that has five versions: the Heligmosomum mixtum allele, the H. bakeri (primary and alternative) version, and the H. polygyrus (primary and alternative) version. For the hyper haplotype to be present since the split, the tree must look like. (Heligmosomum mixtum allele(H.bakeri allele 1, H. polygyrus allele 1)),(H.bakeri allele 2, H. polygyrus allele 2). The readers will feel convinced if a sufficient number of alleles are present in the assemblies.

We respectfully disagree that our gene trees of shared haplotypes must display this topology to show that they were present in the last common ancestor. This is for two important reasons:

1. We sequenced only three *H. bakeri* individuals and two *H. polygyrus* individuals. It's therefore highly likely that we have not sampled all possible hyper-divergent haplotypes circulating in both species.
2. One of the two or more hyper-divergent haplotypes might have been lost in one species (e.g. both haplotypes have been maintained in *H. bakeri*, but only one haplotype maintained in *H. polygyrus*). Both haplotypes do not need to be present in both species for haplotype sharing to be true.

Indeed, previous studies that have identified ancient balancing selection (see *Leffler EM et al. 2013. Multiple instances of ancient balancing selection shared between humans and chimpanzees. Science. 339:1578–1582* and *Koenig D et al. 2019. Long-term balancing selection drives evolution of immunity genes in Capsella. Elife. 8:e43606*) did not show that both haplotypes were present in both species. As in our study, these authors identified regions where one of the two haplotypes in their focal species was more closely related to a comparator species than it was to the alternate haplotype. In our analyses, we found many *H. bakeri* haplotypes that were more closely related to one of the *H. polygyrus* haplotypes than they were to the alternate *H. bakeri* haplotype. Over 40% of these genes had neighbouring genes that also showed evidence of haplotype sharing. Where we studied specific regions in detail, our gene trees were entirely consistent with both read and whole-

genome alignments. We therefore believe that the approach we employed to identify haplotype sharing between *H. bakeri* and *H. polygyrus* is the correct one.

Other comments

L36 The results were not unexpected. There are plenty of literatures that have documented the presence of balancing selection blocks including the ones introduced (for instance Doyle et al 2020) and published by the authors (for instance Stevens et al 2022).

We have re-written this sentence to read *“Together, our results suggest that the selection pressures exerted by the host immune response have played a key role in shaping patterns of genetic diversity in the genomes of parasitic nematodes.”*

*L76-86 This is not true. There are now many reference genomes that are produced outside the Sanger Institute that the authors failed to recognize. Looking at WormBase ParaSite (WBPS) and sort by N50, there are some plant parasitic nematodes that are arguably also very important that have the assembly in chromosomes. Even beyond the database, assembly of *Ascaris suum* (Wang et al 2020 Current Biology) and *H. glycines* (Masonbrink et al 2021 Mol. Ecol. Res) are now in chromosomes also important resources. The authors should acknowledge these efforts and that parasite genomes are being revamped with reference genomes.*

We thank the reviewer for bringing this to our attention. This paragraph was intended to focus on animal parasitic nematodes (e.g. we say that *“mature stages of many parasitic nematodes...are only accessible from the host post-mortem”*) but that was not clear in our writing. We have changed the final sentence in this paragraph to read *“As a result, chromosome-level reference genomes exist for only a handful of animal parasitic nematodes, primarily those of medical or veterinary importance where individuals are large or the species has been inbred”*. We have also added citations for *Ascaris suum* (Wang et al. 2020) and two recently published chromosome-level genomes for *Cylicocyclus nassatus* (Sallé et al. 2023) and *Teladorsagia circumcincta* (Hassan et al. 2023).

L79 The ultra-low Pacbio kit only require 5ng of genomic DNA, which have been routinely sequenced in single insects. The authors should acknowledge this advancement. Note: The ultra-low Pacbio was referred as "similar" protocol in L483. Since the PiMmS could sequence with fewer gDNA, the authors are recommended to discuss in details.

We have edited this sentence to acknowledge the genomes that PacBio ULI has enabled, which now reads *“However, even with these drawbacks, PiMmS and protocols such as the PacBio ultra-low input protocol, which has been used to sequence single arthropods and nematodes^{29,68,69},*

represent an exciting new opportunity for parasite genomics.”. However, because we haven’t empirically tested the input requirements of PiMmS or PacBio ULI, we have opted not to add text to the discussion comparing these protocols.

L125-126 The read N50 from 9.6-11.1kb was quite low to standard HiFi data, which may be due to the original gDNA quality or the PiMmS protocol. Please explain in more details.

As noted in the methods, we intentionally sheared the DNA to ~10 kb. The PCR enzymes used in PiMmS (and in long-range PCR amplification in general) are typically not able to efficiently amplify fragments above 30 kb and so we aim to shear our DNA to a length of 10-15 kb prior to amplification. PacBio ULI libraries have a similar read N50, as the manufacturer also recommends shearing DNA to ~10 kb prior to the ULI protocol (see <https://www.pacb.com/wp-content/uploads/Application-Note-Considerations-for-Using-the-Low-and-Ultra-Low-DNA-Input-Workflows-for-Whole-Genome-Sequencing.pdf>).

Figure 1B It is not known how the cumulative contig plot of 1B shows how the variations in assembly metric were largely driven by read coverage.

We have removed the reference to Figure 1B in that sentence and now only reference Table S1 (which lists both read coverage and assembly metrics).

Figure 1B Type PRJEB15393 in legend.

We thank the reviewer for catching this. The correct BioProject is PRJEB15396, as displayed in Figure 1B, but the caption was incorrect. We have edited the caption to read PRJEB15396.

L176 Please explain source of short-read RNA-seq.

We edited this sentence to read “*we used a combination of evidence from short-read RNA-seq derived from pools of individuals*”. We generated the short-read RNA-seq for *H. polygyrus* as part of this study (described in the ‘Short-read mRNA-seq for *H. polygyrus*’ methods section). We used publicly available RNA-seq data for *H. bakeri*, the source of which is noted in the ‘Protein-coding gene prediction’ subsection in the methods as “*For H. bakeri, we aligned six libraries of publicly available RNA-seq data¹¹⁸ (BioProject PRJNA486010) to the soft-masked nxHelBake1.1 reference genome...*”.

L186 Please explain "due to technical reasons rather than biological differences". Do the 1,505 more genes in *H. polygyrus* were left unannotated in *H. bakeri* due to "technical reasons" or they were false positives?

The sentence that immediately follows provides an explanation for why the gene counts differ: “Relative to *H. bakeri*, the *H. polygyrus* genome has both higher BUSCO completeness (93.7% vs 92.0%) and duplication (2.5% vs 1.2%) scores and the gene set has substantially more single exon genes (3,277 vs 2,406) (Table S2).”. The difference in gene number is for two main reasons: (1) the slightly higher completeness and duplication of the *H. polygyrus* genome relative to the *H. bakeri* genome, meaning more genes are present and/or duplicated in the *H. polygyrus* assembly and (2) more fragmented gene models or false positives in the *H. polygyrus* genome, as indicated by a larger number of single exon genes (which are typically rare in eukaryotic genomes).

Table 1 some details should be spent on describing the almost ~50Mb difference in *H. bakeri* WBPS17 and nxHelBake1.1. Is it to do with phased assembly or PCR bias in the PiMmS?

To determine why there was a 50 Mb difference between the previous versions of the *H. bakeri* reference genome (nHp v2.0) and the new version, we first mapped the PacBio HiFi reads for nxHelBake1 to nHp v2.0 and calculated the average read depth of each contig (see histogram below). In total, 10,623 contigs, spanning 14.4 Mb, had 0x coverage. These contigs were generally short (N50 of 1.7kb). There are 14,770 contigs, spanning 27.2 Mb with a coverage $\leq 5x$. Without doing further analyses, we are unable to say what these contigs are. Based on the publicly available blobtoolkit plot for nHp v2.0 (see https://www.ebi.ac.uk/ena/browser/view/GCA_900096555.1), the assembly does not appear to contain large amounts of contamination. However, we note that ~20 Mb of the assembly does not have a taxonomic annotation and therefore may be derived from unrecognised contaminants. It is also possible that these contigs represent duplicated sequences (e.g. from repeats) that are redundant in the assembly or haplotypes that are present in the laboratory population of *H. bakeri* that we did not sample.

nxHelBake1 PacBio HiFi read coverage of the nHp v2.0 genome assembly.

We also used minimap2 to align the nHp v2.0 contigs to our new reference genome and calculated alignment depth at each position. 56.8 Mb of the new reference genome (~9%) had a depth of two or greater. 112 protein-coding genes were overlapped at least 50% by these apparently duplicated regions, 31 (28%) of which are within hyper-divergent haplotypes. Therefore, it appears that, for some regions of the genome, both hyper-divergent haplotypes are present in the assembly.

In addition to the above analyses, we note that our *H. bakeri* reference genome is marginally less biologically complete than nHp v2.0 (92.0% BUSCO completeness compared with 94.3%). If we assume this difference is representative of the assembly as a whole, this could account for ~17 Mb of the size difference.

We therefore believe that the 50 Mb difference in size is due to a combination of (a) a large number of short contigs in nHp v2.0 that may be redundant, derived from unsampled haplotypes, or derived from unrecognised contaminants, (b) certain regions being present in more than one copy in the nHp v2.0 assembly, including at regions containing hyper-divergent haplotypes, and (c) a slight reduction in completeness in the new *H. bakeri* genome relative to nHp v2.0. Given the complexity associated with these results, we don't believe we could effectively represent them by adding extra rows to Table 1.

L232-234. A figure is needed to show the inversion as well as repeat content across the X chromosome in both species. This will help to understand the apparent lack of points in the middle of X chromosome in Figure 2D and S6A.

We have added a new supplementary figure (Figure S7; which we have pasted below) that shows the inversion and repeat content. Many bins within the apparently inverted regions show 100% repeat content. This new figure is referenced in the main text as “*While we observe an apparent inversion on the X chromosome, the central region of the X is highly repetitive in both genomes (Figure S7) and the order of the contigs within this region in our reference genomes is uncertain.*”.

Figure S7: Repeat content and synteny in the *H. bakeri* and *H. polygyrus* X chromosomes

The relative position for 260 BUSCO genes in *H. bakeri* and *H. polygyrus* X chromosomes are shown as circles coloured by Nigon element. Repeat content in 100 kb windows is shown for both X chromosomes; lines represent LOESS smoothing functions fitted to the data. The location of an apparent inversion is indicated with dotted lines. The inversion-containing regions in both X chromosomes are highly repetitive in both genomes and the order of the contigs within these regions are uncertain in our reference genomes.

L243 The authors have referred that the two species have diverged million of years ago in several places of the manuscript so it would be good to put a rough estimate here, as opposed to refer the *C. briggsae* and *C. nigoni* divergence and left the readers wondering

We agree with the reviewer that having a divergence time estimate for *H. bakeri* and *H. polygyrus* would be extremely valuable. Unfortunately, divergence times are incredibly difficult to estimate in nematodes with any degree of accuracy. The divergence time estimated for *C. briggsae* and *C. nigoni* by Thomas *et al.* (2015) was estimated using the following equation:

$$T = (d_s - \pi_{anc}) / (2\mu) \text{ (Gillespie and Langley 1979)}$$

Where T is equal to the number of generations, d_s is equal to the synonymous site divergence, π_{anc} represents the nucleotide diversity at synonymous sites in the last common ancestor, and μ represents the mutation rate. Thomas *et al.* used the *C. elegans* mutation rate of 2.7×10^{-9} per site per generation (Denver *et al.* 2009) and assumed an average of 10 generations per year (the minimum generation time in *Caenorhabditis* spp. is ~ 3 days).

Obtaining a plausible estimate of the synonymous site diversity in the last common ancestor (π_{anc}) of *H. bakeri* and *H. polygyrus* is difficult because we lack a species-wide resequencing dataset for either species. We therefore estimated synonymous site diversity using the species-wide resequencing dataset available for *H. contortus* (Sallé *et al.* 2018). Using short-read data for 34 isolates that had at least 5x coverage, we estimated synonymous site diversity (π) in 11 populations, which averaged 3.19%.

The generation time in *H. polygyrus* is ~16 days (corresponding to a theoretical maximum number of ~ 23 generations per year). However, as in *Caenorhabditis*, the average number of generations completed each year is likely to be substantially lower than the theoretical maximum. While there are no estimates of the average number of generations per year for any *Heligmosomoides* species in the literature, we found several estimates for related strongylids (largely from parasites of livestock) in the literature that ranged from one generation per year to six generations per year.

Using the equation above with the *H. contortus* synonymous site diversity of 3.19% as our π_{anc} , the *C. elegans* mutation rate, and the synonymous site divergence (d_s) between *H. bakeri* and *H. polygyrus* of 6.62%, we estimated 6.4 million generations since *H. bakeri* and *H. polygyrus* last shared a common ancestor. This corresponds 1.1 million years ago (assuming an average of 6 generations per year) to 6.4 million years ago (assuming an average of 1 generation per year) since *H. bakeri* and *H. polygyrus* last shared a common ancestor. This estimate relies heavily on three assumptions:

1. That present-day *H. contortus* populations are a good proxy for the diversity in the last common ancestor of *H. bakeri* and *H. polygyrus*

2. That 1-6 generations per year is a plausible range for the average number of generations in the *H. bakeri* and *H. polygyrus* lineage
3. That the *C. elegans* mutation rate is similar to that of *Heligmosomoides*

We have added these divergence time estimates to the '*The H. bakeri and H. polygyrus genomes are highly divergent*' results section and added a new methods section titled '*Divergence time estimation*'.

While performing this analysis, we realised that the d_s between the two *Caenorhabditis* sister species pairs was substantially higher than between *H. bakeri* and *H. polygyrus*, which was largely masked by our amino acid identity analysis. We have therefore made the following changes to our manuscript:

- We have replaced the plot showing amino acid identity between *H. bakeri* and *H. polygyrus* in Figure 2D with the average d_s between *H. bakeri* and *H. polygyrus* across the six *H. bakeri* chromosomes
- We have replaced the supplementary plots showing amino acid identity in the three species pairs with new plots that show the average d_s between five nematode sister species pairs: *Brugia malayi* and *B. pahangi*, *Caenorhabditis briggsae* and *C. nigoni*, *Caenorhabditis remanei* and *C. latens*, *H. bakeri* and *H. polygyrus*, and *O. volvulus* and *O. ochengi*. The divergence between *H. bakeri* and *H. polygyrus* is substantially higher than that between the filarial species pairs but is substantially lower than the *Caenorhabditis* species pairs.
- We have updated the relevant results and discussion sections to reflect these new analyses and have removed text related to the amino acid divergence analysis.

Figure S7 C) label should be *H. bakeri* nxHelBake3

We have corrected this figure (Figure S9 in the updated manuscript).

Reviewer 3

Heligmosomoides bakeri and *Heligmosomoides polygyrus* are parasitic nematodes of rodents that are widely used laboratory models because of their phylogenetic proximity to human hook worms. This paper reports chromosome-level assemblies of these two *Heligmosomoides* species. What makes this study valuable is that they used single-worm long-read sequencing to generate assemblies and compare the genomes of multiple individuals of these species, which has not been previously reported in other parasitic nematodes. It should also be noted that they successfully constructed chromosome-level assemblies from single individuals using pooled DNA HiC information. Long-read sequence data from multiple individuals were used for haplotyping and detection of hyper-variable regions. The genome comparison revealed the phylogenetic relationship of the two species, which were previously unclear as to whether or not they were the same species. The haplotype information also showed that there is 'haplotype sharing' between the two species, thought to be the result of blanching selection. These regions have been maintained through long periods of inbreeding in the laboratory and appear to be important for parasitism. The study also suggests that the worms used in laboratories are likely to be genetically different between laboratories, despite having the same origin, and warns that phenotypic analyses can vary between laboratories.

The genome sequencing and informatics analysis were beautifully done and the paper is logical and well written. Unfortunately, the paper does not include any experimentally based data, such as phenotypic observations followed by gene disruption, but I think the novelty described at the beginning of the paper is sufficient to compensate for this.

We thank the reviewer for their positive feedback on our manuscript and we agree that it would be fascinating to explore the functional roles of the hyper-divergent haplotypes we discovered in *H. bakeri* in future.

Minor points;

Line 168; 124,254 and 45,333 what are these numbers for? Probably they are full length transcripts of *Heligmosomoides bakeri* and *Heligmosomoides polygyrus*, respectively. Also any idea why the numbers are so different from each other.

As noted in the text, these are the number of full-length transcripts that were assembled from the long-read cDNA data for each species. *H. bakeri* has far more assembled transcripts than *H. polygyrus* because we generated far more cDNA data for *H. bakeri*, which we note in the text (“We sequenced the resulting cDNA libraries using PacBio Sequel IIe platform and generated 3.6 and 0.7 Gb of PacBio HiFi data per individual”).

Line472; “contigs” instead of “assemblies”?

We have replaced ‘assemblies’ with ‘contigs’.

Line500-505; redundant with the introduction. Please reduce text amount.

We have reduced this text into one sentence.

Line542; that have previously reported to be associated with?

We have edited the sentence to read “...*that have previously been reported to be associated with parasite biology...*”.

Additional changes

We have made a set of minor text changes that we believe improve the clarity of the writing but do not alter the content of the manuscript. These changes, in addition to all changes in response to the reviewers' comments, are indicated in the word document tracked changes.

In our discussion section about hyper-divergent haplotypes, we have added a citation to a preprint by Cole *et al.* (see <https://www.biorxiv.org/content/10.1101/2021.05.26.445462v3>), wherein they find that the most diverse regions of the *Strongyloides ratti* genome are enriched for genes expressed in the parasitic life-stage.

In our discussion section about the species status of *H. bakeri* and *H. polygyrus*, we have added a citation to a recently published analysis by Musah-Eroje *et al.* that shows that the two species differ morphologically (in their spicule length).

In the original version of our manuscript, Figure 1A showed labels with family names but these were incorrectly applied. We have replaced these with suborder names in the revised version of the manuscript.

REVIEWERS' COMMENTS

Reviewer #1 (Remarks to the Author):

The authors have thoughtfully addressed my previous comments, elaborated on the haplotype filtering and validation methods, added divergence time estimations, and further improved the manuscript's clarity. The paper is in good shape and can be recommended for publication in Nature Communications.

Reviewer #2 (Remarks to the Author):

I am happy with the revised manuscript. One last minor suggestion would be to publicise the assembly and annotation in Wormbase Parasite!

Response to reviewers

Reviewer comment

Author response

Reviewer 1

The authors have thoughtfully addressed my previous comments, elaborated on the haplotype filtering and validation methods, added divergence time estimations, and further improved the manuscript's clarity. The paper is in good shape and can be recommended for publication in Nature Communications.

We thank the reviewer for their time and effort during the review process.

Reviewer 2

I am happy with the revised manuscript. One last minor suggestion would be to publicise the assembly and annotation in Wormbase Parasite!

Both the nxHelBake1.1 and ngHelPoly1.1 assemblies and annotations will be released in WormBase ParaSite release WBPS20. We thank the reviewer for their time and effort during the review process.